# Causal-JEPA: Learning World Models through Object-Level Latent Masking

Heejeong Nam [1]  Quentin Le Lidec [2] [*]  Lucas Maes [3] [*]  Yann LeCun [2]  Randall Balestriero [1]

## Abstract

World models require robust relational understanding to support prediction, reasoning, and control. While object-centric representations provide a useful abstraction, they are not sufficient to capture interaction-dependent dynamics. We therefore propose C-JEPA, a simple and flexible object-centric world model that extends masked joint embedding prediction from image patches to object-centric representations. By masking object-level latents and requiring each masked object state to be inferred from the surrounding context, C-JEPA imposes structured partial observability during training, creating counterfactual-like prediction queries that discourage shortcut solutions and make interaction-dependent prediction necessary under the learning objective. Empirically, C-JEPA leads to consistent gains in visual question answering, with an absolute improvement of about 20% in counterfactual reasoning over the same architecture without object-level masking. On agent control tasks, C-JEPA enables substantially more efficient planning by using only 1% of the total latent input features required by patch-based world models, while achieving comparable performance. Finally, we provide a formal analysis demonstrating that object-level masking induces useful inductive bias by controlling observability. Our code is available at https://github.com/galilai-group/cjepa.

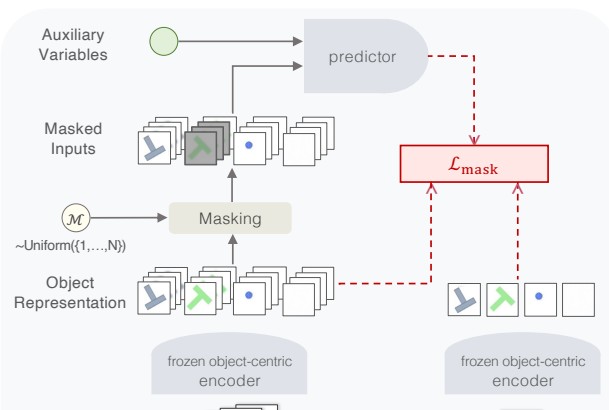

*Figure 1.* C-JEPA training pipeline. A frozen encoder extracts object-centric representations, followed by selective masking across history. The predictor recovers masked history slots and predicts future latent states, conditioned on optional auxiliary variables, via a joint masked-history and forward-prediction objective.

## 1. Introduction

World models (Ha & Schmidhuber, 2018) provide a unifying abstraction for learning, predicting, and reasoning about the dynamics of complex environments, enabling scalable planning and control directly in latent space from high-dimensional observations (Hafner et al., 2019; 2025). At their core, effective world modeling benefits from representations that expose entities and their interactions, avoiding reliance on spurious pixel-level correlations (Zholus et al., 2022; Nishimoto & Matsubara, 2025; Ferraro et al., 2023; Feng et al., 2025; Lei et al., 2025). Inspired by this perspective, object-centric models have been extensively developed for learning visual dynamics (Kipf et al., 2022; Zadaianchuk et al., 2023; Wu et al., 2023; Mosbach et al., 2025; Villar-Corrales et al., 2023), and more recently adopted as a foundation for learning world models (Ferraro et al., 2023; Nishimoto & Matsubara, 2025). However, learning on top of object-centric representations alone is not sufficient. Existing works have shown that, without explicit mechanisms to guide interaction learning, models can easily fall back on object self-dynamics or exploit incidental correlations (Villar-Corrales et al., 2023; Lei et al., 2025). This realization has spurred growing interest in object-centric world models that explicitly learn interactions (Kipf et al., 2020; Feng et al.,

[*]Equal contribution  [1]Brown University, GalilAI [2]New York University [3]Mila & Université de Montréal. Correspondence to: Heejeong Nam <heejeong_nam@brown.edu>, Lucas Maes <lucas.maes@mila.quebec>.

*Proceedings of the $43^{rd}$ International Conference on Machine Learning*, Seoul, South Korea. PMLR 306, 2026. Copyright 2026 by the author(s).

2025; Mosbach et al., 2025) or causal structure (Zholus et al., 2022; Nishimoto & Matsubara, 2025; Lei et al., 2025). These approaches often enforce interactions by separating temporal dynamics from object interactions (Mosbach et al., 2025; Villar-Corrales et al., 2023; Feng et al., 2025), regularizing attention sparsity (Lei et al., 2025), leveraging graph structures (Kipf et al., 2020), or relying on downstream-specific methods (Ferraro et al., 2023; Nishimoto & Matsubara, 2025). Together, these works emphasize the importance of interactions in world modeling, while leaving open the question of how interaction structure can be made functionally necessary through the learning objective itself.

Accordingly, we propose Causal-JEPA (C-JEPA), a simple yet effective approach for object-centric world modeling with a causal inductive bias induced by object-level masking interventions on predictor observability. This design is motivated by the fact that existing patch-based masked prediction approaches (He et al., 2022; Fan et al., 2023; Thoker et al., 2025; Tessler et al., 2024; Tong et al., 2022) optimize for local patch correlations, without enforcing object-level interaction reasoning. We propose an object-level masking that trains the model to infer an object's latent trajectory from others' evolving states, inducing counterfactual-like interventions during learning. Our masking strategy prevents shortcut solutions such as trivial temporal interpolation, making interaction reasoning necessary for minimizing the prediction objective. At the same time, C-JEPA remains architecturally flexible and can incorporate auxiliary variables such as actions and proprioception.

Notably, the compactness of object-centric representations, when combined with the Joint Embedding Predictive Architecture (JEPA) (LeCun, 2022), encourages the model to concentrate on the essential factors governing object dynamics within a low-dimensional latent space, thereby reducing both computational and memory overhead. In contrast to patch-based predictors, where attention scales quadratically with a large number of patch tokens, entity-level representations apply the same quadratic attention mechanism over a much smaller token set, resulting in significantly lower training and inference costs.

In Sec. 5, we evaluate C-JEPA from two perspectives, visual reasoning and predictive control. We first evaluate our approach on CLEVRER (Yi et al., 2020), a video QA benchmark involving multi-object interactions, and show that object-level masking leads to consistent improvements. In particular, C-JEPA improves visual question answering, with a roughly 20% absolute gain on counterfactual reasoning over the same architecture without object-level masking. We further evaluate C-JEPA in a model predictive control (MPC) setting on the Push-T manipulation task. Compared to patch-based world models such as DINO-WM (Zhou et al., 2025), C-JEPA achieves comparable task performance

while using only 1.02% of the total input feature size, which translates to more than $8\times$ faster model predictive control. Finally, in Sec. 6, we formally characterize object-level masked prediction as a inductive bias, showing that object masking make interaction reasoning necessary for minimizing the training objective. Taken together, this leads to four main contributions:

- We introduce C-JEPA, an object-centric world model that uses object-level masking as a latent intervention, encouraging interaction-dependent prediction.

- C-JEPA consistently improves visual reasoning over multi-object interactions, with particularly large gains on counterfactual questions.

- C-JEPA enables highly efficient predictive control, achieving comparable performance to patch-based world models while using orders-of-magnitude fewer tokens and substantially faster MPC planning.

- We provide an analysis of how object-level masked prediction can induce a causal inductive bias, making interaction reasoning functionally necessary.

## 2. Related Works

### 2.1. Structured World Models for Dynamics Learning

World modeling in dynamical scenes requires structured representations capturing entities and their interactions. Some approaches rely on weak supervision over entity information (Zholus et al., 2022; Ferraro et al., 2023; Lei et al., 2025) or reinforcement-learning–specific architectures (Ferraro et al., 2023; Nishimoto & Matsubara, 2025), while others build self-supervised and generally applicable world models using object-centric representations. For example, several methods explicitly decompose self-dynamics and interactions through architectural factorization (Feng et al., 2025; Villar-Corrales et al., 2023; Mosbach et al., 2025). Subsequent work move beyond reconstruction to reduce redundancy in learned representations. While reconstruction-free, SlotFormer (Wu et al., 2023) does not explicitly enforce interaction structure, and C-SWM (Kipf et al., 2020) relies on a fixed relational graph, limiting adaptability. SPARTAN (Lei et al., 2025) learns world models directly in latent space and encourages interaction selectivity via sparse attention. However, none of these works induces interaction-aware structure through the learning objective itself.

### 2.2. Masking-Based Representation Learning

Masked image modeling was originally introduced as a scalable self-supervised learning paradigm to improve representation quality (He et al., 2022; Tong et al., 2022). Subsequent work explored guided masking strategies to enhance

representations by selectively masking regions informed by motion or dynamics (Fan et al., 2023; Thoker et al., 2025; Tessler et al., 2024). Nakano et al. (2024) combines masking with object-centric representations, but applies masking to image patch tokens for computational efficiency, while object representations are used only as conditioning signals. Masking has also been used in causal discovery to identify latent structure (Ng et al.; Yang et al., 2021; Nam, 2023), but these methods typically assume fixed or identifiable graphs and focus on disentanglement or structure recovery, rather than flexible world modeling. Finally, (Xiao et al., 2023) interpreted masking as inducing counterfactual variations, which aligns with our use of masking, but they also assumed a structural causal model and focused on robust fine-tuning. In contrast, our work adopts masking as an object-level latent intervention to shape predictive dependencies in a world model as a principled inductive bias for learning dynamics.

## 3. Preliminaries

**Joint Embedding Predictive Architecture** JEPA (Le-Cun, 2022) defines a self-supervised learning paradigm that learns to predict in representation space without reconstructing pixels, focusing instead on modeling predictive relationships between latent embeddings that capture semantic structure in the data. This formulation has been progressively extended from image-level masked joint embedding prediction in I-JEPA (Assran et al., 2023), to spatiotemporal tube-based prediction in video with V-JEPA (Bardes et al., 2024), and further to integrated understanding, prediction, and planning in V-JEPA2 (Assran et al., 2025). By avoiding reconstruction-driven objectives, JEPA-style models learn representations that are directly aligned with prediction and control, making them particularly well suited for world modeling and autonomous decision-making (LeCun, 2022; Assran et al., 2023; Bardes et al., 2024; Assran et al., 2025). Following prior works (Zhou et al., 2025; Li et al., 2025), we employ frozen target encoders and optimize the predictor with a joint embedding prediction objective, aligning predicted latents to target latents.

**Object-Centric Representation via Slot Attention** Slot Attention (Locatello et al., 2020) was introduced as a mechanism for learning object-centric representations by iteratively grouping feature maps into a fixed set of slots through competitive attention. By alternating between attention-based assignment and slot updates, it produces a set of object-level representations without requiring supervision. Subsequent work extended this formulation from images to videos (Kipf et al., 2022; Zadaianchuk et al., 2023; Aydemir et al., 2023; Singh et al., 2022). Unlike image-based Slot Attention, video extensions must additionally maintain slot identity over time, which is typically achieved by conditioning slot updates on previous-frame slots. Notably,

some approaches such as VideoSAUR (Zadaianchuk et al., 2023) leverage powerful pretrained foundation models, including DINO (Oquab et al., 2024) and DINOv2 (Caron et al., 2021), as feature encoders, inheriting strong semantic and invariance properties from large-scale pretraining. In our framework, we use Slot Attention to encode each observation into a fixed-size set of object-centric latent states, yielding a set representation $S_t = \{s_t^1, \ldots, s_t^N\}$, where each slot corresponds to a distinct entity in the scene.

## 4. Method

In this section, we introduce C-JEPA, an object-centric latent world model that applies object-level masking during training to induce interaction-aware predictive representations. All notation is summarized in Appendix A.

### 4.1. Problem Setting

We consider a video-based dynamical system observed through a sequence of images. Let $X_t \in \mathbb{R}^{H \times W \times C}$ denote the pixel-level observation at time $t$. An object-centric encoder $g$ maps each frame to a set of object-centric slots,

$$S_t = g(X_t) = \{s_t^1, \ldots, s_t^N\}, \qquad s_t^i \in \mathbb{R}^d, \qquad (1)$$

where $N$ is the fixed number of slots and $d$ is the dimensionality of each slot. The set representations $S_t$ are permutation-equivariant with respect to the slot ordering. We consider a history window of length $T_h$ and a prediction horizon of length $T_p$. Accordingly, we define the history index set $T := \{t - T_h + 1, \ldots, t\}$, and the full history–future prediction interval $\mathcal{T} := \{t - T_h + 1, \ldots, t + T_p\}$. Given past object states $S_T$, the world model aims to predict future object states $\{S_{t+1}, \ldots, S_{t+T_p}\}$.

**Masked and Visible Object Sets.** For any time step $\tau$ in the history window $T$, we partition the object set $S_\tau$ into a masked subset and an unmasked context subset. Let $\mathcal{M}_\tau \subset \{1, \ldots, N\}$ denote the indices of masked objects at time $\tau$. We then define

$$S_\tau^{\mathrm{m}} = \{s_\tau^i \mid i \in \mathcal{M}_\tau\}, \qquad S_\tau^{\mathrm{c}} = \{s_\tau^j \mid j \notin \mathcal{M}_\tau\}. \quad (2)$$

The masked set $S_\tau^{\mathrm{m}}$ denotes the subset of object states selected for masking, whose observations are replaced by mask tokens $\tilde{S}_\tau^{\mathrm{m}}$, while the remaining states $S_\tau^{\mathrm{c}}$ are provided as context. During training, $f$ learns to recover masked tokens within $S_T$ and to predict future states $\{S_{t+1}, \ldots, S_{t+T_p}\}$ under masking. At inference time, $f$ is used solely for forward prediction from a fully observed history $S_T$, without masking.

**Auxiliary Observable Variables.** In addition to object states, we allow for auxiliary observable variables that influence state transitions. For example, when actions and

*Figure 2.* Object-level latent masking in C-JEPA. Selected object slots are masked across time, except for a minimal identity anchor, forcing the predictor to infer object dynamics from interactions with other objects and auxiliary variables.

proprioceptive signals are available, we denote these variables by $U_t = \{a_t, p_t\}$, where $a_t$ represents actions and $p_t$ represents proprioceptive signals. These variables are treated as external to the object-level state representation and incorporated as additional conditioning inputs alongside object states. This enables the model to capture both object–object dependencies and action-mediated influences within a unified object-centric framework, consistent with treating observable sources as auxiliaries (Kim et al., 2025).

### 4.2. Causal-JEPA

We now describe the core design of C-JEPA, which introduces object-level masking for learning interaction-aware dynamics.

**Object-Level Masking for Learning Interaction.** Fig. 2 illustrates our object-level masking. At each time step $t$, the object-centric representation $S_t$, together with auxiliary variables $U_t$, is represented as a set of entity tokens $Z_t = \{S_t, U_t\}$, which serves as the input to the predictor.

While future entity tokens are always masked for prediction, we additionally mask the observable latent states of selected objects across the history window $T$ according to a masking index set $\mathcal{M}$, preserving only the earliest time step $t_0$ as an identity anchor. The identity anchor is introduced to address the permutation equivariance of slot-based representations with respect to object ordering, which motivates omitting positional encodings along the entity dimension, consistent with prior work (Wu et al., 2023; 2021). As a result, identity information of each entity must be provided for masked tokens, enabling the transformer predictor to distinguish *which* entities are masked and to predict appropriately.

Concretely, we define the masked token as

$$\tilde{z}_\tau^i = \phi(z_{t_0}^i) + e_\tau, \qquad (3)$$

where $\phi$ is a linear projection, $z_{t_0}^i$ serves as an identity

anchor, and $e_\tau$ is a learnable embedding combined with temporal positional encoding. This masking operation can be interpreted as a latent-level control of observability, forcing prediction to rely on interactions with other entities. We elaborate more on this interpretation in Appendix B.1.

**Learning Objective.** C-JEPA serves as the latent predictor in the object-centric world model, operating on masked object representations to learn dynamics. Fig. 1 illustrates the overall architecture and training procedure of C-JEPA. The predictor $f$ is a ViT-style masked transformer with bidirectional attention (Assran et al., 2023; Bardes et al., 2024; Assran et al., 2025), enabling joint inference over masked object tokens across the history window and future horizon. Here, $\bar{Z}_\mathcal{T}$ denotes the masked input sequence in which selected entity tokens $z$ are replaced by mask tokens $\tilde{z}$ across both the history window and the future horizon (Fig. 1). The predictor outputs latent predictions as

$$\hat{Z}_\mathcal{T} = f(\bar{Z}_\mathcal{T}). \qquad (4)$$

Training minimizes a masked latent prediction objective by predicting all masked object tokens over the interval $\mathcal{T}$:

$$\mathcal{L}_{\text{mask}} = \mathbb{E}\left[\sum_{\tau \in \mathcal{T}} \sum_{i=1}^{N} \mathbf{1}\left[\bar{z}_\tau^i \neq z_\tau^i\right] \left\|\hat{z}_\tau^i - z_\tau^i\right\|_2^2\right], \qquad (5)$$

where the indicator $\mathbf{1}[\bar{z}_\tau^i \neq z_\tau^i]$ selects all tokens masked in the input over $\mathcal{T}$. This masked latent prediction objective can be viewed as a combination of mask reconstruction over the history window and prediction over the future horizon,

$$
\begin{aligned}
\mathcal{L}_{\text{mask}} &= \underbrace{\mathbb{E}\left[\left\|\hat{z}_\tau^i - z_\tau^i\right\|_2^2 \,\Big|\, i \in \mathcal{M},\, \tau \leq t\right]}_{\mathcal{L}_{\text{history}}} \\
&+ \underbrace{\mathbb{E}\left[\left\|\hat{Z}_\tau - Z_\tau\right\|_2^2 \,\Big|\, \tau > t\right]}_{\mathcal{L}_{\text{future}}}.
\end{aligned} \qquad (6)
$$

The history term suppresses reliance on trivial self-dynamics under partial observability, while the future term enforces alignment with forward world modeling. Together, object-level masking makes interaction reasoning functionally necessary for minimizing the predictive objective.

**Remark 1** (Causal Interpretation of Object-Level Masking). *In C-JEPA, prediction is governed by structured dependencies among object states and exogenous variables, without assuming a fixed or explicitly parameterized causal graph. Here, the term "causal" refers to temporally directed predictive dependencies from past object states to future states that remain stable under object-level masking, rather than claims of causal identifiability.*

*Object-level masking acts as a latent intervention on predictor observability that poses counterfactual-like queries*

*during training by selectively removing access to object states. This introduces a causal inductive bias that enforces reliance on interaction-relevant variables. In line with observations in Lei et al. (2025), the predictor's attention mechanism gives rise to state-dependent interaction patterns that can be interpreted as a soft, local relational structure capturing predictive influence. We formalize this interpretation in Sec. 6.*

**Inference.** At inference time, C-JEPA performs forward latent prediction following Eq. 4, with a fully observable history and masking applied only to future tokens. This procedure aligns with the standard objective of latent world modeling, enabling direct rollout of future object states for downstream planning, control, and reasoning.

## 5. Experiments

In this section, we evaluate C-JEPA from two complementary perspectives, focusing on visual reasoning in Sec. 5.1 and predictive control in Sec. 5.2. Across experiments, we primarily adopt VideoSAUR (Zadaianchuk et al., 2023) as the object-centric encoder $g$, which aggregates frozen DINOv2 (Oquab et al., 2024) features into object-centric latents. In Sec. 5.1, we additionally report results using SAVi (Kipf et al., 2022) to ensure comparability with baseline models. Implementation and pretraining details are deferred to Appendix D.

### 5.1. Causal-JEPA for Visual Reasoning

**Task and Evaluation.** We first evaluate visual reasoning on CLEVRER (Yi et al., 2020), a synthetic video benchmark for physical and causal understanding in dynamic scenes. CLEVRER consists of videos depicting multiple interacting objects and question–answer pairs for evaluating descriptive, predictive, explanatory and counterfactual reasoning over object interactions.

To enable reasoning from predicted object trajectories, we adopt ALOE (Ding et al., 2021), following the evaluation setup of SlotFormer (Wu et al., 2023). ALOE is designed to perform reasoning directly on stacked object-centric representations over time, using a language transformer to condition attention on the input question. Briefly, we train C-JEPA and baseline world models, and roll out 128-frame input videos to 160 frames to produce imagined trajectories. ALOE is then trained on these model-generated trajectories for downstream questions. We use a fixed number of seven object slots across all CLEVRER experiments. Further data and evaluation details are in Appendix F.

**Baseline Models.** We evaluate three distinct object-centric world modeling architectures. Since ALOE presupposes object-centric representations, VQA evaluation is restricted

*Table 1.* VQA accuracy under varying numbers of masked objects. Overall accuracy is reported per question, with counterfactual accuracy additionally reported per option and per question. (V) uses VideoSAUR and (S) uses SAVi encoders. Performance changes relative to the unmasked baseline are indicated by arrows.

| Model | $\|\mathcal{M}\|$ | Average per que. (%) | Counterfactual VQA (%) | |
| --- | --- | --- | --- | --- |
| | | | per opt. | per que. |
| OC-JEPA (V) | 0 | 82.79 | 79.53 | 47.68 |
| C-JEPA (V) | 1 | 83.95 ↑ 1.16 | 80.34 ↑ 0.81 | 49.67 ↑ 1.99 |
| | 2 | 84.56 ↑ 1.77 | 80.61 ↑ 1.08 | 50.25 ↑ 2.57 |
| | 3 | 87.61 ↑ 4.82 | 86.49 ↑ 6.96 | 63.60 ↑ 15.92 |
| | 4 | **89.40 ↑ 6.61** | **88.67 ↑ 9.14** | **68.81 ↑ 21.13** |
| OC-JEPA (S) | 0 | 77.28 | 76.69 | 41.10 |
| C-JEPA (S) | 1 | 78.39 ↑ 1.11 | 77.26 ↑ 0.57 | 43.13 ↑ 2.03 |
| | 2 | **83.88 ↑ 6.60** | **85.16 ↑ 8.47** | **60.19 ↑ 19.09** |
| | 3 | 79.02 ↑ 1.74 | 77.78 ↑ 1.09 | 43.77 ↑ 2.67 |
| | 4 | 73.28 ↓ 4.00 | 73.55 ↓ 3.14 | 34.06 ↓ 7.04 |

to object-centric models, and non-object-centric baselines are omitted. To ensure a fair comparison under the same reconstruction-free setting as C-JEPA, we additionally evaluate reconstruction-free variants of the baseline models.

- **SlotFormer** (Wu et al., 2023) performs autoregressive rollouts over object latents, serving as a canonical baseline without explicit interaction constraints. We evaluate both *SlotFormer (+/–Recon.)*.

- **OCVP-Seq** (Villar-Corrales et al., 2023) factorizes self-dynamics and object interactions at the attention level. We evaluate both *OCVP-Seq (+/–Recon.)*.

- **OC-JEPA** is a history-unmasked variant of C-JEPA that uses the same architecture while masking only future tokens, isolating the effect of masking-induced inductive bias.

All baselines use the same pretrained SAVi encoder for fair comparison. We defer additional details to Appendix H.

**Results.** Tab. 1 shows the comparison between OC-JEPA and C-JEPA provides a clean ablation demonstrating that performance gains stem from the learning objective rather than object-centric representations alone. Introducing object-level masking consistently improves performance, indicating that the causal inductive bias induced by the objective is the key driver. Notably, improvements are substantially larger on counterfactual questions than on overall accuracy or other question categories (See full results in Appendix I), suggesting that masking does not merely enhance predictive accuracy but directly strengthens counterfactual reasoning. This aligns with the training setup, where

*Table 2.* Baseline comparison using a SAVi encoder. For models with and without reconstruction, arrows indicate differences relative to the reconstruction-based variant.

| Model | Average per que. (%) | Counterfactual VQA (%) | |
|---|---|---|---|
| | | per opt. | per que. |
| SlotFormer | 79.44 | 79.28 | 47.29 |
| *(-) recon. loss* | 44.94 ↓ 34.50 | 55.62 ↓ 23.66 | 11.10 ↓ 36.19 |
| OCVP-Seq | 83.11 | 83.21 | 56.06 |
| *(-) recon. loss* | 80.09 ↓ 3.02 | 77.46 ↓ 5.75 | 43.00 ↓ 13.06 |
| OC-JEPA | 77.28 | 76.69 | 41.10 |
| C-JEPA | **83.88** | **85.16** | **60.19** |

*Table 3.* Push-T planning success rates across world models and token budgets. Changes are reported relative to OC-DINO-WM, and $|\mathcal{M}| = 1$ is used for C-JEPA.

| # Token $\times d$ | Model | | Success Rate (%) |
|---|---|---|---|
| $196 \times 384$ | DINO-WM | | **91.33** |
| | DINO-WM-Reg. | | 88.00 |
| $6 \times 128$ | OC-DINO-WM | (ref.) | 60.67 |
| | OC-JEPA | (+JEPA) | 76.00 ↑ +15.33 |
| | C-JEPA | (+Mask) | 88.67 ↑ **+28.00** |

object-level masking exposes the model to counterfactual-like queries through controlling observabilitly. Finally, excessive masking can remove informative dependencies, indicating an optimal masking regime that depends on the robustness of the underlying object representations from the encoder.

Tab. 2 shows a comparison against object-centric baselines under reconstruction and non-reconstruction settings. Results show that removing reconstruction leads to a severe performance degradation in SlotFormer (Wu et al., 2023), indicating a strong reliance on pixel-level supervision. OCVP-Seq (Villar-Corrales et al., 2023) exhibits a comparatively modest decline, suggesting that its architectural factorization partially mitigates the loss of reconstruction-based signals. In contrast, C-JEPA achieves the strongest performance without any reconstruction, highlighting the effectiveness of its decoder-free design and masking-based learning objective. Full results are available in Appendix I.

## 5.2. Causal-JEPA for Efficient World Modeling

**Task and Evaluation.** To evaluate world model learning in a control-oriented setting, we use the PushT (Chi et al., 2025) environment, a robotic manipulation benchmark involving contact-rich object interactions in which an agent must reason about object dynamics and contact effects over time to achieve goal-directed behavior. Following the planning pipeline introduced in Zhou et al. (2025), at time step $t$ planning is performed by solving a finite-horizon optimal control problem in the latent object-centric state space induced by the learned world model. Given the current history window $S_T$, we optimize a sequence of future actions $a_{t:t+H-1}$ over a planning horizon $H$ as

$$a^*_{t:t+H-1} \triangleq \arg\min_{a_{t:t+H-1}} \left\| \hat{S}_{t+H} - S_g \right\|_2^2, \qquad (7)$$

where $S_g$ denotes the latent representation of the desired goal state. This objective encourages plans whose long-term predicted outcome matches the desired goal, without imposing intermediate costs. The optimization prob-

lem in Eq. equation 7 is solved using the Cross-Entropy Method (CEM). Planning is performed repeatedly in a model-predictive control (MPC) fashion as new observations become available. A task is considered successful when the distance between the actual system state and the desired goal state in the original (non-latent) state space falls below a predefined threshold. We use a fixed number of four object slots across all Push-T experiments. Further details are in Appendix G.

**Baseline Models.** We construct a series of world model baselines based on DINO-WM and compare them to VideoSAUR-based C-JEPA. This allows all models, including C-JEPA, to start from the same frozen DINOv2 embeddings and differ only in the predictor architecture and its inputs, enabling a controlled comparison. We defer additional details to Appendix H.

- **DINO-WM** (Zhou et al., 2025) operates on patch-level representations and employs a autoregressive causal transformer predictor.

- **DINO-WM (reg.)** (Darcet et al., 2024) uses a DINOv2-with-register backbone with the same patch-based predictor, isolating the effect of representation.

- **OC-DINO-WM** applies an object-centric encoder on top of DINOv2 patch embeddings, while keeping the DINO-WM predictor and objective unchanged, isolating the effect of representation alone.

- **OC-JEPA** is a history-unmasked variant of C-JEPA, masking only future tokens.

**Results.** Table 3 highlights a clear progression from patch-based to object-centric world modeling. The patch-based DINO-WM achieves the highest success rate, but at the cost of operating over a large token space. Replacing patch tokens with object-centric slots in OC-DINO-WM reduces the latent token space to *1.02%*, but results in a pronounced performance degradation, showing that object-centric representations alone do not suffice for accurate long-horizon prediction. Introducing a JEPA-style predictor in OC-JEPA partially recovers performance while preserving the efficiency

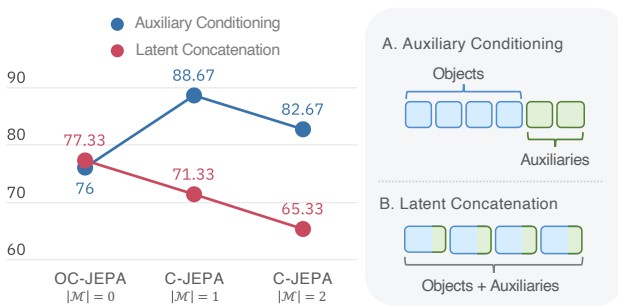

*Figure 3.* Comparison of auxiliary variable integration methods.

of object-centric representations, indicating the advantage of joint latent prediction over autoregressive objectives.

Finally, C-JEPA closes the performance gap with DINO-WM (Zhou et al., 2025) by introducing object-level masking during training. Despite using only *1.02%* of the latent tokens, it achieves performance comparable to patch-based world models. Because predictor rollouts dominate the computational cost in model-based planning, this reduction directly translates to efficiency gains. Under identical settings on a single L40s GPU, C-JEPA achieves over $8\times$ faster planning, requiring 673 seconds on average across three seeds to evaluate 50 trajectories, compared to 5,763 seconds for DINO-WM. Overall, C-JEPA reconciles the efficiency of object-centric world models with the performance of image-based approaches by combining joint latent prediction with object-level masking.

**Action and Proprioception as Auxiliary Nodes.** As discussed in Sec. 4.1, C-JEPA treats auxiliary variables, such as actions and proprioceptive signals, as separate entities that condition the predictor. In contrast, DINO-WM (Zhou et al., 2025) concatenates these signals with patch representations, incorporating control inputs into the visual representation. As shown in Fig. 3, modeling auxiliary variables as separate entities consistently outperforms concatenation into object latents, justifying their treatment as explicit variables.

### 5.3. Further Analysis of the Masking Objective

**Qualitative analysis on PHYRE.** To further examine whether the masking objective remains useful in more complex physical scenes, we additionally conduct qualitative case studies on PHYRE (Bakhtin et al., 2019). Following the perspective of Lei et al. (2025), we use cross-slot attention patterns in the predictor as a proxy for local temporal interaction structure, and analyze them together with imagined rollout trajectories. As shown in Appendix J, C-JEPA produces more physically plausible rollouts than its unmasked counterpart OC-JEPA in representative multi-body interaction cases, and its attention patterns are more concentrated on physically relevant slots. These results provide qualitative evidence that object-level masking encourages more

interaction-aware predictive dependencies in temporally extended physical environments.

**Ablation on Masking Strategy.** In addition to object-level masking, we evaluate token- and tube-level masking across both reasoning and control tasks. While token- or tube-level masking can occasionally match the performance of object-level masking depending on the task, object-level masking provides a more structured and controllable inductive bias. This results in more stable training behavior and superior performance on interaction-heavy queries. We provide a comprehensive comparative analysis in Appendix K.

## 6. Theoretical Perspective: Causal Inductive Bias of Latent Masking

This section analyzes why C-JEPA yields stronger forward prediction than future-only world modeling objectives.

### 6.1. Assumption

We first set practically motivated assumptions that clarify the scope of our analysis. Detailed discussion of the assumptions is provided in Appendix L.

**Assumption 1** (Temporally Directed Predictive Dependencies). *Object-level state transitions are governed by time-directed predictive dependencies. The future state of an object is determined by past object observations and auxiliary variables without requiring instantaneous causal effects within the same time step.*

**Assumption 2** (Shared Transition Mechanism). *Latent state transitions are governed by a shared mechanism. Although object states vary across time and episodes, the conditional distribution of future states given a finite history remains invariant across trajectories.*

**Assumption 3** (Object-Aligned Latent Representation). *Each slot corresponds to a coherent object-level state variable and these representations provide a sufficient abstraction for reasoning about object dynamics.*

**Assumption 4** (Finite-History Sufficiency). *A finite history window of length $T_h$ suffices to predict future object states under the predictive dependencies of the system.*

**Note.** We emphasize that we do not assume causal sufficiency. It is important to permit unobserved confounders when using object-centric representations, as they do not assume full observability of the underlying system. We also do not assume a first-order Markov process at the object level. Future object states may depend on a history of past states rather than solely on $Z_t$, as velocity cannot be inferred from a single observation. Furthermore, we do not assume global sparsity of the underlying dynamics, which has been shown to be overly restrictive (Pandaram et al., 2025).

## 6.2. Interaction-Inducing Inductive Bias

This subsection analyzes the inductive bias induced by predicting missing object representations from partial observations over a temporal window $T$. We show that masked history prediction forces the predictor to rely on a minimal sufficient set of contextual variables beyond the target object itself, which we formalize as an *influence neighborhood*.

**Definition 1** (Influence Neighborhood under Masked Completion). *Let $Z_T$ denote the set of entity tokens over the history window, and let $Z_T^{(-i)}$ denote all variables in $Z_T$ excluding the history of object $i$, except for a minimal identity anchor used to preserve object identity. For a masked object state $z_t^i$ at time $t$, the* influence neighborhood $\mathcal{N}_t(i)$ *is a minimal sufficient subset $\mathcal{N}_t(i) \subseteq Z_T^{(-i)}$ such that*

$$p\left(z_t^i \mid Z_T^{(-i)}\right) = p\left(z_t^i \mid \mathcal{N}_t(i)\right). \tag{8}$$

*Minimality is defined in the sense that no strict subset of $\mathcal{N}_t(i)$ satisfies this condition.*

The influence neighborhood captures the smallest set of contextual variables that must be consulted in order to recover the state of a masked object under partial observability. The concept of influence neighborhoods is closely related to notions of Markov blankets and sufficient statistics (Pearl, 2009; Koller & Friedman, 2009), and we discuss these connections in Appendix B.2. We define the influence neighborhood for a single masked object state, and masking multiple objects during training corresponds to jointly learning several such conditional dependencies in parallel, without altering the definition. The following result formalizes the basic predictive consequence of object-level masking. It should be interpreted as a statement about predictive sufficiency under masking, rather than as a claim of recovering true causal interactions.

**Theorem 1** (Interaction Necessity under Masked History Completion). *Consider the masked history prediction loss from Eq. 6 for object $i$ at time $t$:*

$$\mathcal{L}_{\text{history}}^i := \mathbb{E}\left[\left\|\hat{z}_t^i - z_t^i\right\|_2^2\right], \tag{9}$$

*where $\hat{z}_t^i$ is computed from the observable history $Z_T^{(-i)}$ following Eq. 4, and the expectation is taken with respect to the conditional distribution of $z_t^i$ given $Z_T^{(-i)}$. Under Assumptions 1–4 and Definition 1, the optimal predictor that minimizes equation 9 satisfies*

$$\hat{z}_t^{i\,*} = \mathbb{E}\left[z_t^i \mid Z_T^{(-i)}\right] = \mathbb{E}\left[z_t^i \mid \mathcal{N}_t(i)\right]. \tag{10}$$

*Consequently, any predictor that fails to utilize information in $\mathcal{N}_t(i)$ cannot attain the minimum achievable expected reconstruction error under masked completion.*

As masked $z_t^i$ is not directly observable, the optimal predictor must rely on other variables to reduce uncertainty. By Definition 1, $\mathcal{N}_t(i)$ is the minimal sufficient subset that preserves the conditional distribution of $z_t^i$. Any predictor that ignores variables in $\mathcal{N}_t(i)$ therefore incurs strictly higher expected loss. A complete proof is provided in Appendix M.1.

**Corollary 1** (Learning Intervention-Stable Influence Neighborhoods). *Optimizing $\mathcal{L}_{\text{mask}}$ under repeated exposure to diverse object-level masking encourages state-dependent attention patterns that align with the influence neighborhood $\mathcal{N}_t(i)$. This can be interpreted as a soft, local relational structure capturing predictive influence.*

Corollary 1 is closely related to prior work on invariant causal prediction (Peters et al., 2016) and invariant risk minimization (Arjovsky et al., 2020), which study predictive dependencies that remain stable across masking or environments. In this sense, object-level masking in C-JEPA can be viewed as inducing a collection of latent interventions on observability of the predictor, under which intervention-stable influence neighborhoods emerge. Further discussion is provided in Appendix M.4.

**Remark 2** (Influence Neighborhoods as Predictive Sets). *Influence neighborhoods provide a practical alternative to full causal discovery in complex dynamical systems. In realistic settings with latent confounders and high-dimensional state spaces, identifying true causal parents may be infeasible or ill-defined (Seitzer et al., 2021; Ke et al., 2021). Importantly, we interpret influence neighborhoods as predictively sufficient sets under masking, rather than as estimates of true causal parents or causal mechanisms, since they may include variables that are informative under latent confounding or partial observability.*

In this sense, intervention-stable influence neighborhoods capture variables that are useful for prediction under controlled perturbations, making them relevant for reasoning, planning, and model-based control. C-JEPA encourages predictors to rely on such neighborhoods through its learning objective, without requiring an explicit causal graph.

**Remark 3** (Transfer of Bidirectional Training to Forward Prediction). *Due to bidirectional masked prediction during training, the influence neighborhood $\mathcal{N}_t(i)$ abstracts away temporal direction. Although the true system dynamics evolve via a forward transition $s_{t+1} = \Psi(s_t)$, $\mathcal{N}_t(i)$ can be interpreted as a direction-agnostic interaction structure that abstracts away edge direction and captures variables jointly informative about an object's state, independent of whether this information arises from past or future observations.*

*As a result, minimizing $\mathcal{L}_{\text{mask}}$ induces latent representations that encode interaction constraints invariant to the direction of information flow. Under Assumption 2, predicting an object's state during training depends on the same variables*

*that govern its forward dynamics from past observations, irrespective of temporal direction.*

## 7. Conclusion

In this paper, we presented C-JEPA, which combines joint embedding prediction with object-level masking to introduce the causal inductive bias directly through the learning objective. To our knowledge, this is the first work to integrate JEPA with object-centric world modeling. By treating object masking as counterfactual-like query, C-JEPA learns interaction-aware dynamics without relying on reconstruction losses or task-specific supervision. Empirically, C-JEPA yields strong gains in visual reasoning, with especially large improvements on counterfactual questions. In predictive control, it enables highly efficient planning with orders of magnitude fewer tokens than patch-based world models, while achieving comparable performance.

We also identify several limitations. Performance depends on the quality of the object-centric encoder, which can limit the performance ceiling. Moreover, while we formally characterize influence neighborhoods, we do not directly validate them on datasets with explicit temporal causal graphs, leaving this to future work. Another promising direction is jointly refining object-centric encoders using strong pre-trained backbones without representational collapse (Đukić et al., 2025), as well as evaluating C-JEPA in more complex environments with richer interactions. Overall, this work establishes object-level masking as a principled and effective mechanism for introducing causal inductive bias into efficient, interaction-aware world models.

## Impact Statement

This paper presents work whose goal is to improve the efficiency and interaction-aware learning of task-agnostic world models and their applications. While advances in world modeling may have broad downstream impacts in areas such as robotics, simulation, and decision-making systems, the methods proposed here are primarily foundational and methodological. We do not foresee any immediate negative societal consequences specific to this work beyond those generally associated with the deployment of learned models, and therefore do not highlight any particular impacts here.

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

# A. Notations

*Table A1.* Notations used throughout the paper.

| Notation | Description |
|---|---|
| $X_t \in \mathbb{R}^{H \times W \times C}$ | Pixel-level image frame observed at time $t$ |
| $g$ | Object-centric encoder mapping pixels to slot representations |
| $f$ | Latent predictor mapping past states to future states |
| $\phi$ | Linear projector used to construct masked object tokens |
| $S_t = \{s_t^1, \ldots, s_t^N\}, s_t^i \in \mathbb{R}^d$ | Set of object-centric slot representations at time $t$ |
| $N$ | Fixed number of object slots |
| $d$ | Dimensionality of each slot representation |
| $T_h, T_p$ | History window length and prediction horizon length respectively |
| $T = \{t - T_h + 1, ..., t\}$ | History index set |
| $\mathcal{T} = \{t - T_h + 1, ..., t + T_p\}$ | Full history–future prediction interval |
| $U_t$ | Set of exogenous variables at time $t$ |
| $a_t$ | Action taken at time $t$ |
| $p_t$ | Proprioceptive signal at time $t$ |
| $\mathcal{M} \subset \{1, \ldots, N\}$ | Index set of masked objects |
| $e_\tau$ | Learnable embedding combined with temporal positional encoding |
| $S_t^{\mathrm{m}} = \{s_t^i \mid i \in \mathcal{M}\}$ | Set of masked object states at time $t$ |
| $S_t^{\mathrm{c}} = S_t \setminus S_t^{\mathrm{m}}$ | Set of unmasked object states |
| $Z_t = \{S_t, U_t\}$ | Set of entity tokens at time $t$ |
| $\tilde{z}_t^i$ | Masked token for object $i$ at time $t$ |
| $\bar{z}_t^i$ | Slot token after applying object-level masking |
| $\bar{Z}_t$ | Set of slot tokens after masking at time $t$ |
| $\mathcal{Z}_t^{(-i)}$ | Available context variables excluding object $i$ at time $t$ |
| $\mathcal{Z}_{t,\mathrm{mask}}^{(-i)}$ | Masked context variables under object-level masking |
| $\mathcal{N}_t(i)$ | Influence neighborhood of object $i$ (single-step form) |
| $\hat{z}_{t+1}^i$ | Predicted future state of object $i$ at time $t+1$ |
| $\hat{z}_{t+1:t+T_p}$ | Predicted future object states over the horizon |
| $\mathbf{1}[\,\cdot\,]$ | Indicator function (equals 1 if condition holds, 0 otherwise) |

# B. Discussion

## B.1. Interpretation of Object-Level Masking as a Latent Intervention on Predictor Observability

We clarify how object-level masking in C-JEPA can be interpreted as a latent intervention on observability. Crucially, masking alters the information available to the predictor without modifying the underlying data-generating or transition mechanism.

**Lemma 1** (Latent Intervention via Object-Level Masking). *Object-level masking removes direct access to the current latent state of a masked object $z_t^i$ while preserving access to other object states and auxiliary variables. As a result, prediction of the future object state is conditioned on a restricted set of observable variables,*

$$\bar{Z}_t = \left( Z_t \setminus \{z_t^i\} \right) \cup \{\tilde{z}_t^i\},$$

*where $\tilde{z}_t^i$ is a masked token that does not reveal the current value of $z_t^i$. This intervention restricts observability at time $t$ while leaving the transition dynamics unchanged.*

**Discussion.** At a given time step, object-level masking replaces the current latent state of a target object with a masked token, thereby restricting the predictor's access to that object while leaving all other inputs unchanged.

From this perspective, object-level masking acts as a *latent intervention on observability*. Rather than intervening on the data-generating process itself, masking induces counterfactual-like prediction problems by forcing the model to reason about future states under controlled information removal. This interpretation aligns with viewing masking as an intervention in latent space, enabling the learning of interaction-dependent predictive structure without requiring multiple environments or explicit causal graph manipulation.

### B.2. Relation to Classical Causal Concepts

The notion of an *influence neighborhood* is related to several classical concepts in causal inference and probabilistic modeling, but is deliberately defined in a weaker and more operational sense, tailored to masked prediction in object-centric world models.

**Relation to Markov blankets and causal parents.** In probabilistic graphical models (Koller & Friedman, 2009), the Markov blanket of a variable is the minimal set of variables that renders it conditionally independent of all others. Similarly, in structural causal models (Pearl, 2009), the causal parents of a variable represent its direct causes. Both notions aim to characterize *minimal sufficient sets* under strong assumptions, including access to a correct causal graph, causal sufficiency, and full observability.

In contrast, influence neighborhoods are defined through the masked completion task in Definition 1. They characterize the minimal subset of *observable variables* within $Z_T^{(-i)}$ that are sufficient to predict a masked object state $z_t^i$ when its own history is unobserved, except for a minimal identity anchor. No assumptions are made about the identifiability of a true causal graph.

As a consequence, an influence neighborhood need not coincide with the set of causal parents. It may include variables that are causally downstream, correlated through latent confounders, or otherwise informative under partial observability. Influence neighborhoods should therefore be interpreted as *predictively sufficient* sets under masking, rather than as estimates of true causal mechanisms. Nevertheless, we argue that influence neighborhoods still introduce a causal inductive bias, in the sense that forward prediction in the world model is constrained to be temporally directed, thereby favoring one-way predictive dependencies from past to future rather than arbitrary symmetric associations.

**Relation to invariant causal prediction and invariant risk minimization.** Invariant causal prediction (ICP) (Peters et al., 2016) and invariant risk minimization (IRM) (Arjovsky et al., 2020) aim to identify predictive relationships that remain stable across environments or interventions. This emphasis on stability aligns conceptually with our focus on predictive dependencies that persist under object-level masking.

However, existing methods typically assume access to multiple environments from data-generating process or externally specified interventions. In contrast, C-JEPA induces systematic variations in observability through object-level masking within a single dataset. Influence neighborhoods are thus defined relative to *latent observability interventions*, rather than invariance across externally labeled environments or do-interventions on the data-generating process.

**Summary.** Overall, influence neighborhoods generalize classical notions of minimal predictive sets to object-centric, temporally extended, and partially observed settings. They capture which variables are necessary for prediction under masked completion, without assuming causal sufficiency or aiming to recover a fixed causal graph. This perspective enables principled analysis of interaction structure in realistic world models while remaining agnostic to causal identifiability.

## C. Dataset

### C.1. CLEVRER (Yi et al., 2020)

CLEVRER consists of videos with 128 frames at a spatial resolution of $480 \times 320$. The dataset is split into 10,000 training videos, 5,000 validation videos, and 5,000 test videos.

The CLEVRER VQA benchmark includes four categories of questions: descriptive, counterfactual, explanatory, and predictive. Descriptive questions are evaluated per question, whereas the remaining categories are evaluated both per question and per answer option.

Due to current unavailability of the evaluation server, we report results on the validation set using the same evaluation

protocol as the test set. All models are trained exclusively on the training split, and the validation set is treated as a held-out test set during evaluation. No validation data is used for training or model selection. The subset of test-set results available to us exhibits trends consistent with those reported on the validation set.

The number of objects present in a scene is not fixed and may change over time, as objects can enter or leave the scene during a video. Across the dataset, the maximum number of simultaneously visible objects is six. Accordingly, we use a fixed set of seven object slots throughout, where one slot implicitly captures background or empty regions.

### C.2. Push-T (Chi et al., 2025)

The Push-T dataset consists of videos with variable temporal length, where each frame is rendered at a resolution of $224 \times 224$. The training set contains 18,410 trajectories, and the validation set has 21 trajectories.

The task involves a simple planar manipulation scenario in which a controllable agent, represented as a blue circle, must push a green T-shaped object into a target configuration, a gray T-shaped object. Success is defined by placing the green object within a predefined distance threshold of the goal pose.

Proprioceptive inputs have dimensionality four, and actions have dimensionality two. Each scene consistently contains three objects, and we therefore use four object-centric slots, including one slot reserved for background.

## D. Encoders

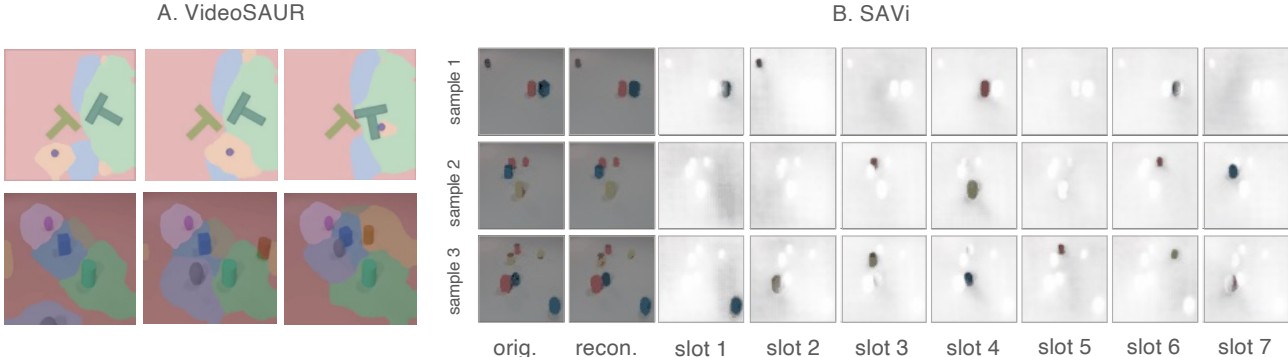

*Figure A1.* Slot visualization from object-centric encoders.

### D.1. VideoSAUR (Zadaianchuk et al., 2023)

VideoSAUR (Zadaianchuk et al., 2023) is a video slot attention model built on a frozen DINOv2 (Oquab et al., 2024) backbone that aggregates patch-level features into object-centric slots using a SAVi-style grouping mechanism (Kipf et al., 2022). It additionally employs a temporal similarity loss to encourage consistency across time, and we adopt it as our primary object-centric encoder.

**Common Setup.** Across all experiments, VideoSAUR is trained following the standard protocol. Visual features are extracted using a frozen DINOv2 ViT-S/14 backbone, producing 196 patch tokens per frame with dimensionality 384. Patch features are projected to a 128-dimensional latent space and grouped into object-centric slots using Slot Attention with two iterations. The DINOv2 backbone remains frozen throughout training. Models are trained for 100k steps using Adam with a base learning rate of $1 \times 10^{-4}$, scaled linearly by batch size, and an exponential decay schedule with 2k warmup steps. Gradients are clipped at 0.05. The training objective combines a feature reconstruction loss on DINOv2 features and a temporal similarity loss. Unless otherwise specified, remaining training details follow the VideoSAUR setup used for YouTube-VIS by Wu et al. (2023).

**Training Details on CLEVRER.** For CLEVRER, frames are temporally subsampled with stride two. Training uses short clips of six frames, while validation uses clips of length ten. We use $N = 7$ slots, a batch size of 32, and set the temporal similarity loss weight to 0.3. Data normalization on ImageNet statistics and random horizontal flips are applied.

**Training Details on Push-T.** For Push-T, we use clips of six frames with stride two for both training and validation, $N = 4$ slots, and a batch size of 64. The temporal similarity loss weight is increased to 0.25. Training is performed on a Push-T dataset variant, where each object is moving in a random manner to increase visual and dynamical diversity.

### D.2. SAVi (Kipf et al., 2022)

SAVi (Kipf et al., 2022) is a video slot attention model that learns object-centric representations by grouping frame-level visual features into a fixed set of slots using iterative attention. It reconstructs input frames from slot representations and serves as a standard object-centric encoder in prior world model and video reasoning work.

**Training Details on CLEVRER.** For experiments requiring a SAVi-based encoder, we use the stochastic SAVi model following SlotFormer (Wu et al., 2023), adopting the same architecture and training configuration. Training is performed on six-frame video clips resized to $64 \times 64$. The encoder maps RGB frames to 128-dimensional features using a convolutional backbone, followed by Slot Attention with $N = 7$ slots and two iterations. Each slot has dimensionality 128. We use the stochastic SAVi formulation with a Gaussian prior of variance 0.01. Models are trained for 8 epochs using Adam with a learning rate of $1 \times 10^{-4}$ and a cosine schedule with linear warmup over the first $2.5\%$ of steps. Gradients are clipped at 0.05. The batch size is 32 for training and 64 for validation. The objective consists of an image reconstruction loss and a KL regularization term weighted by $1 \times 10^{-4}$.

### D.3. Auxiliary Variable Encoders.

If available, actions and proprioceptive signals are treated as auxiliary variables and embedded separately from object-centric latents. Both are encoded using lightweight temporal embedders implemented as one-dimensional convolutional layers over the temporal axis. Each embedder maps raw inputs to a 128-dimensional embedding space for C-JEPA, OC-JEPA, and OC-DINO-WM, while using a 10-dimensional embedding space for DINO-WM and DINO-WM with Register.

## E. Predictors

### E.1. Design Choice

We adopt a ViT-style masked predictor rather than an autoregressive Transformer for modeling object-centric dynamics. In object-centric representations, object states do not evolve as independent first-order Markov processes, but are shaped by interactions that may span multiple time steps. Autoregressive predictors impose a sequential dependency structure that conditions each prediction on previously generated tokens, which can bias learning toward local self-dynamics. In contrast, masked prediction allows the model to attend jointly to the entire history window and infer masked object states in parallel. This architectural choice aligns naturally with the C-JEPA objective, which requires reasoning over interaction-dependent context rather than sequential token generation.

### E.2. Implementation Details

We maintain object latent states with slot dimensionality 128 throughout the whole experiment. Future prediction is performed by a Transformer-based predictor with six layers, 16 attention heads, head dimension 64, and an MLP hidden dimension of 2048. Our framework is built on top of `stable-pretraining` (Balestriero et al., 2025) and `stable-worldmodel` (Maes et al., 2026).

### E.3. Training Details

For Push-T, the model operates on a temporal history window of size three, with frames subsampled using a frame skip of five. At each time step, the model predicts a single future latent state. For CLEVRER, the model operates on a temporal history window of size six, with frames subsampled using a frame skip of two, predicting ten future latent states. During training, object-level masking is applied by randomly masking between zero and two object slots for Push-T and between zero and four object slots for CLEVRER.

The model is trained for 30 epochs using the Adam optimizer with a batch size of 256. The predictor, action encoder, and proprioceptive encoder use a learning rate of $5 \times 10^{-4}$. All experiments are conducted on a single GPU, on pre-extracted object embeddings.

# F. Visual Question Answering

### F.1. ALOE (Ding et al., 2021)

For visual question answering, we adopt ALOE (Ding et al., 2021), a transformer-based model for reasoning over object-centric video representations. ALOE operates directly on object-centric representations extracted from video frames. Given a sequence of object embeddings over time and a natural-language question, ALOE encodes the question using a language transformer and represents object trajectories as temporally ordered tokens. A stack of attention layers then performs joint reasoning over objects, time steps, and question tokens, allowing information to be dynamically routed based on the query. Importantly, ALOE does not rely on explicit symbolic programs or predefined causal graphs. Instead, relational, temporal, and counterfactual reasoning emerge from attention over object-level embeddings conditioned on the question.

**Training Details on CLEVRER.** We train ALOE for visual question answering on the CLEVRER dataset using object-centric slot trajectories extracted by the encoder. We follow SlotFormer (Wu et al., 2023) for the training configuration. Each training sample consists of object slot embeddings over 25 time steps, together with a natural-language question and multiple answer choices. We use a maximum of 6 objects per scene, with each slot represented by a 128-dimensional feature vector. Object order is fixed across time to preserve temporal consistency.

Questions and answer choices are tokenized with maximum lengths of 20 and 12, respectively. Object trajectories, question tokens, and answer tokens are concatenated into a single token sequence and processed by a transformer encoder with 12 layers, 8 attention heads, and a feedforward dimension of 512. Learnable positional embeddings are used. The transformer operates on a shared embedding space of dimension 16, followed by an MLP classifier with hidden size 128 for answer prediction. The model is trained using the Adam optimizer with an initial learning rate of $10^{-3}$ and a cosine decay schedule with linear warmup over the first $10\%$ of training steps. Training is performed for 400 epochs following SlotFormer.

# G. Model Predictive Control Planning Details

We describe the implementation details of the model predictive control (MPC) pipeline used for goal-conditioned planning with learned world models. Planning is performed in the latent space induced by the learned world model. Given the current observation and a goal observation, the planner optimizes a sequence of future actions by rolling out the world model and minimizing a goal-conditioned cost. Action optimization is carried out using the Cross-Entropy Method (CEM), and execution follows a receding-horizon MPC scheme.

**Planning Configuration.** In all experiments, we set the planning horizon to $H = 5$ and use an action block size of $B = 5$, resulting in a total action sequence length of $L = H \times B = 25$. The receding horizon is set equal to the planning horizon ($R = 5$), such that the full optimized action sequence is executed before replanning. The evaluation budget is fixed to 50 environment steps per episode, and the goal observation is defined as the state occurring 25 steps after the initial frame sampled from the dataset.

**Cost Function.** The planning objective minimizes the cost between the predicted last state and the goal state in the latent space. The cost used is the $l_2$-loss. Analogous costs are computed for prospective embeddings, which are then added to the original cost. A rollout is considered successful if the final position error is less than 20 (within a workspace range of $(0, 450)$) and the final orientation error is smaller than $\pi/9$.

**Action Optimization via CEM.** Action sequences are optimized using the Cross-Entropy Method. At each replanning step, CEM samples 300 candidate action sequences from a Gaussian distribution and evaluates their quality from the cost predicted by the world model. At each step, the 30 candidates with the lowest cost are kept as elites to update the Gaussian parameters (mean and variance) used for action sampling. The optimization procedure is repeated for 30 iterations, and the final mean of the distribution is used as the optimized action sequence.

**Hungarian Matching for Object-Centric Models.** For object-centric world models, the ordering of object slots is not often perfectly guaranteed to be consistent across time steps or between predicted and goal states. To ensure a meaningful comparison between object-centric latent states, we apply Hungarian matching when computing rollout trajectories and planning costs. Specifically, at each predicted time step, object slots are matched to goal slots by minimizing the pairwise $\ell_2$ distance in latent space. The planning cost is then computed after alignment, using the matched object representations.

This matching is applied consistently for all object-centric models during MPC planning and does not affect patch-based baselines, where token ordering is fixed.

## H. Baselines

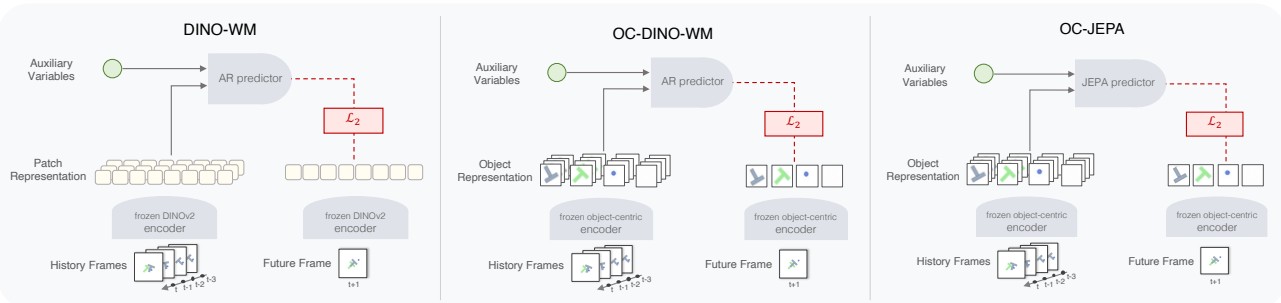

*Figure A2.* Diagram of DINO-WM (Zhou et al., 2025), OC-DINO-WM, and OC-JEPA

**SlotFormer (Wu et al., 2023)** We reproduce SlotFormer on CLEVRER using the best-performing configuration reported in the original work. Training is performed on object-centric slots extracted by a pretrained SAVi (Kipf et al., 2022) encoder. Each training sample consists of 16 frames in total, including 6 burn-in frames and 10 rollout frames, with temporal subsampling by a factor of two. The dynamics model is implemented as a Transformer-based rollouter with four layers, eight attention heads, and a model dimension of 256. Sinusoidal temporal positional encodings are applied, while no slot positional encodings are used.

Training is conducted for 30 epochs (approximately 450k steps) using the Adam optimizer with a learning rate of $2 \times 10^{-4}$. A cosine learning-rate schedule with linear warmup over the first $5\%$ of training steps is applied. No weight decay or gradient clipping is used. We use batch size 32 per GPU, using two Nvidia RTX A6000 GPUs. The default training objective includes both slot-level reconstruction loss and image reconstruction loss, each weighted equally. Image reconstruction is performed using a CNN decoder initialized from a pretrained SAVi checkpoint. All images are resized to a resolution of $64 \times 64$.

**OCVP-Seq (Villar-Corrales et al., 2023)** We reproduce OCVP on CLEVRER using the official implementation and training protocol. OCVP is also trained on top of pretrained SAVi (Kipf et al., 2022) slot representations, which are kept fixed throughout training. Video clips are constructed from CLEVRER sequences resized to a spatial resolution of $64 \times 64$, with frames temporally subsampled by a factor of two. Each training clip consists of 6 context frames followed by 10 future frames to predict, resulting in a total sequence length of 16 frames.

OCVP-Seq implemented as a Transformer with 2 layers, 4 attention heads, and a hidden dimension of 256. The internal token dimension is set to 128, and residual connections are used around the predictor. Autoregressive prediction is performed with an input buffer size of 6 slots. The model is trained for 30 epochs using the AdamW optimizer with a learning rate of $1 \times 10^{-4}$ and no weight decay. A cosine learning-rate schedule with a warmup of 5 epochs is applied, and the minimum learning rate is set to $1 \times 10^{-6}$. Training uses a batch size of 64 and mixed-precision on 3 Nvidia RTX A6000 GPUs. The training objective consists of a mean squared error loss on predicted slot representations. Validation is performed at the end of every epoch, and all reported results follow the same evaluation protocol as other object-centric baselines.

**DINO-WM (Zhou et al., 2025) and DINO-WM with Register (Darcet et al., 2024)** We train DINO-WM and its register-augmented variant on the Push-T manipulation task following the standard DINO-WM training protocol. Both models share the same architecture and hyperparameters, differing only in the use of register tokens during visual encoding for DINO-WM with Register.

Input frames are tokenized into non-overlapping $16 \times 16$ patches. Frames are temporally subsampled with a stride of five. Each training sequence consists of a history window of three frames followed by one prediction step, resulting in a total sequence length of four frames. Visual observations are encoded using a frozen DINOv2 (Oquab et al., 2024) backbone, producing patch-level embeddings that are passed to an autoregressive Transformer-based predictor. The predictor consists

*Table A2.* VQA accuracy under varying numbers of masked objects. (V) uses VideoSAUR and (S) uses SAVi encoders. Performance changes relative to the unmasked baseline are indicated by arrows.

| Model | $|\mathcal{M}|$ | Average per que. (%) | Counterfactual (%) | | Explanatory (%) | | Predictive (%) | | Descriptive (%) |
|---|---|---|---|---|---|---|---|---|---|
| | | | per opt. | per que. | per opt. | per que. | per opt. | per que. | |
| OC-JEPA (V) | 0 | 82.79 | 79.53 | 47.68 | 92.88 | 80.58 | 86.15 | 75.04 | 89.59 |
| C-JEPA (V) | 1 | 83.95 ↑1.16 | 80.34 ↑0.81 | 49.67 ↑1.99 | 93.41 ↑0.53 | 82.22 ↑1.64 | 88.28 ↑2.13 | 78.58 ↑3.54 | 90.39 ↑0.80 |
| | 2 | 84.56 ↑1.77 | 80.61 ↑1.08 | 50.25 ↑2.57 | 93.53 ↑0.65 | 82.54 ↑1.96 | 88.68 ↑2.53 | 79.56 ↑4.52 | 91.02 ↑1.43 |
| | 3 | 87.61 ↑4.82 | 86.49 ↑6.96 | 63.60 ↑15.92 | 95.35 ↑2.47 | 87.30 ↑6.72 | 91.16 ↑5.01 | 83.86 ↑8.82 | 91.98 ↑2.39 |
| | 4 | **89.40** ↑6.61 | **88.67** ↑9.14 | **68.81** ↑21.13 | **96.62** ↑3.74 | **90.74** ↑10.16 | **93.03** ↑6.88 | **86.93** ↑11.89 | **92.84** ↑3.25 |
| OC-JEPA (S) | 0 | 77.28 | 76.69 | 41.10 | 91.20 | 76.04 | 83.48 | 70.51 | 84.05 |
| C-JEPA (S) | 1 | 78.39 ↑1.11 | 77.26 ↑0.57 | 43.13 ↑2.03 | 91.49 ↑0.29 | 77.14 ↑1.10 | 83.67 ↑0.19 | 70.19 ↓0.32 | 85.09 ↑1.04 |
| | **2** | **83.88** ↑6.60 | **85.16** ↑8.47 | **60.19** ↑19.09 | **95.34** ↑4.14 | **87.27** ↑11.23 | **87.46** ↑3.98 | **77.25** ↑6.74 | **87.81** ↑3.76 |
| | 3 | 79.02 ↑1.74 | 77.78 ↑1.09 | 43.77 ↑2.67 | 92.51 ↑1.31 | 79.94 ↑3.90 | 81.42 ↓2.06 | 67.18 ↓3.33 | 85.62 ↑1.57 |
| | 4 | 73.28 ↓4.00 | 73.55 ↓3.14 | 34.06 ↓7.04 | 89.85 ↓1.35 | 72.72 ↓3.32 | 77.23 ↓6.25 | 59.89 ↓10.62 | 80.88 ↓3.17 |

of 6 Transformer layers with 16 attention heads, a feedforward dimension of 2048, and a per-head dimensionality of 64. Dropout with rate 0.1 is applied within the predictor, while embedding dropout is disabled. Actions and proprioceptive signals are embedded using separate encoders as described in Appendix D.3 with embedding dimensionality 10 and provided as auxiliary inputs to the predictor. Models are trained for 10 epochs as training more epochs did not improve the performance and often degrades the performance. We use a batch size of 64. Optimization is performed using Adam with a learning rate of $5 \times 10^{-4}$ for the predictor, action encoder, and proprioceptive encoder. All experiments use a fixed random seed of 42. Unless otherwise specified, all remaining settings follow the original DINO-WM implementation.

**OC-DINO-WM** OC-DINO-WM replaces the patch-based visual encoder of DINO-WM with an object-centric encoder based on VideoSAUR. VideoSAUR also operates on a frozen DINOv2 ViT-S/14 backbone, ensuring compatibility with DINO-WM in terms of visual representation. All other components, including the predictor architecture, auxiliary input handling, and training protocol, are kept identical to DINO-WM. The model is trained for 30 epochs on Push-T.

**OC-JEPA** OC-JEPA is a variant of C-JEPA with history masking disabled ($|\mathcal{M}| = 0$). This baseline isolates the effect of object-centric representations from that of the masking objective, allowing us to assess whether the performance gains of C-JEPA arise from object-centric encoding alone or from object-level history masking. OC-JEPA shares the same architecture and training setup as C-JEPA, differing only in the masking configuration. Details of the predictor architecture and masking scheme are provided in Appendix E.

# I. Full Results

Tab. A2 shows VQA performance under different numbers of masked objects. Introducing object-level masking consistently improves performance over the unmasked baseline, with especially large gains on counterfactual questions. These results indicate that masking object histories provides a meaningful training signal and strengthens interaction-dependent reasoning.

Tab. A3 compares object-centric baselines using a shared SAVi encoder. C-JEPA achieves the best overall performance across all question categories, with particularly strong gains on counterfactual and predictive questions. Notably, unlike prior approaches, C-JEPA attains these improvements without relying on reconstruction losses, suggesting that object-level masked prediction provides a more effective supervisory signal for interaction reasoning.

# J. Qualitative Analysis on PHYRE

We conduct additional qualitative case studies on PHYRE (Bakhtin et al., 2019) to examine whether object-level masking remains useful in more complex physical environments. PHYRE requires reasoning about temporally extended physical dependencies, including gravity, momentum transfer, and multi-body collisions. Unlike CLEVRER, PHYRE does not provide ground-truth temporal local causal graphs, and therefore we do not evaluate causal graph recovery. Instead, following

*Table A3.* Baseline comparison using a SAVi encoder. For models with and without reconstruction, arrows indicate differences relative to the reconstruction-based variant.

| Model | Average per que. (%) | Counterfactual (%) | | Explanatory (%) | | Predictive (%) | | Descriptive (%) |
|---|---|---|---|---|---|---|---|---|
| | | per opt. | per que. | per opt. | per que. | per opt. | per que. | |
| SlotFormer | 79.44 | 79.28 | 47.29 | 92.84 | 80.96 | 83.66 | 70.79 | 85.22 |
| *(-) recon. loss* | 44.94 ↓ 34.50 | 55.62 ↓ 23.66 | 11.10 ↓ 36.19 | 67.70 ↓ 25.14 | 27.89 ↓ 53.07 | 52.52 ↓ 31.14 | 23.83 ↓ 46.96 | 54.67 ↓ 30.55 |
| OCVP-Seq | 83.11 | 83.21 | 56.06 | 94.38 | 85.37 | 86.34 | 75.52 | **87.85** |
| *(-) recon. loss* | 80.09 ↓ 3.02 | 77.46 ↓ 5.75 | 43.00 ↓ 13.06 | 92.70 ↓ 1.68 | 80.38 ↓ 4.99 | 80.72 ↓ 5.62 | 66.08 ↓ 9.44 | 87.24 ↓ 0.61 |
| OC-JEPA | 77.28 | 76.69 | 41.10 | 91.20 | 76.04 | 83.48 | 70.51 | 84.05 |
| C-JEPA | **83.88** | **85.16** | **60.19** | **95.34** | **87.27** | **87.46** | **77.25** | 87.81 |

the perspective of SPARTAN (Lei et al., 2025), we analyze the predictor's cross-slot attention patterns as a qualitative proxy for local temporal interaction structure. We complement this analysis with imagined rollout trajectories to assess whether the learned world model produces physically plausible futures.

**Rollout plausibility.** We first compare imagined rollouts produced by C-JEPA and OC-JEPA, where OC-JEPA denotes the same architecture trained without object-level masking over the history. Representative examples are shown in Fig. A3. Across these cases, OC-JEPA more often produces physically implausible rollouts, suggesting that the unmasked objective can fail to propagate interaction effects over time. For example, when an early collision should influence the subsequent motion or fall of another object, OC-JEPA may fail to reflect this dependency in the imagined future. In contrast, C-JEPA better preserves the qualitative consequences of object interactions. Even when the predicted future does not exactly match the ground-truth trajectory, the rollout remains more physically plausible, indicating that object-level masking helps the predictor rely on contextual physical dependencies rather than only object self-dynamics.

**Attention-based dependency proxy.** We next analyze cross-slot attention patterns in the predictor. Because ground-truth temporal local causal graphs are unavailable in PHYRE, these attention maps should not be interpreted as recovered causal graphs. Rather, they provide a qualitative view of which object slots the predictor consults when predicting a target slot under partial observability. Following SPARTAN (Lei et al., 2025), we use this cross-slot dependency pattern as a proxy for local temporal interaction structure.

Fig. A4 shows a representative case in which the presence of a particular object slot is important for predicting the target slot. In this example, OC-JEPA assigns attention more diffusely and places relatively high attention on less relevant slots. By contrast, C-JEPA concentrates attention more sharply on the slot visually involved in the relevant interaction, indicating that the masked objective encourages the predictor to use interaction-relevant context when recovering masked object states. This behavior is consistent with the role of object-level masking: by removing direct access to selected object histories during training, the predictor is encouraged to infer missing object states from other objects whose states are informative under the physical dynamics.

**Discussion.** These qualitative observations suggest that object-level masking can remain useful beyond the primary benchmark settings considered in the main text. In PHYRE, C-JEPA produces more physically plausible imagined rollouts and exhibits sharper, more interpretable cross-slot attention patterns than its unmasked counterpart. This analysis should be interpreted as evidence of interaction-aware predictive dependency modeling, rather than as evidence of formal causal graph identification.

# K. Exploring Different Masking Strategies

We consider three masking strategies that differ in the structural unit being masked.

- **Object-level masking.** Entire object-centric slots are masked, requiring the model to infer the masked object's latent trajectory from the remaining objects. The masking budget is specified as the number of masked objects out of seven.

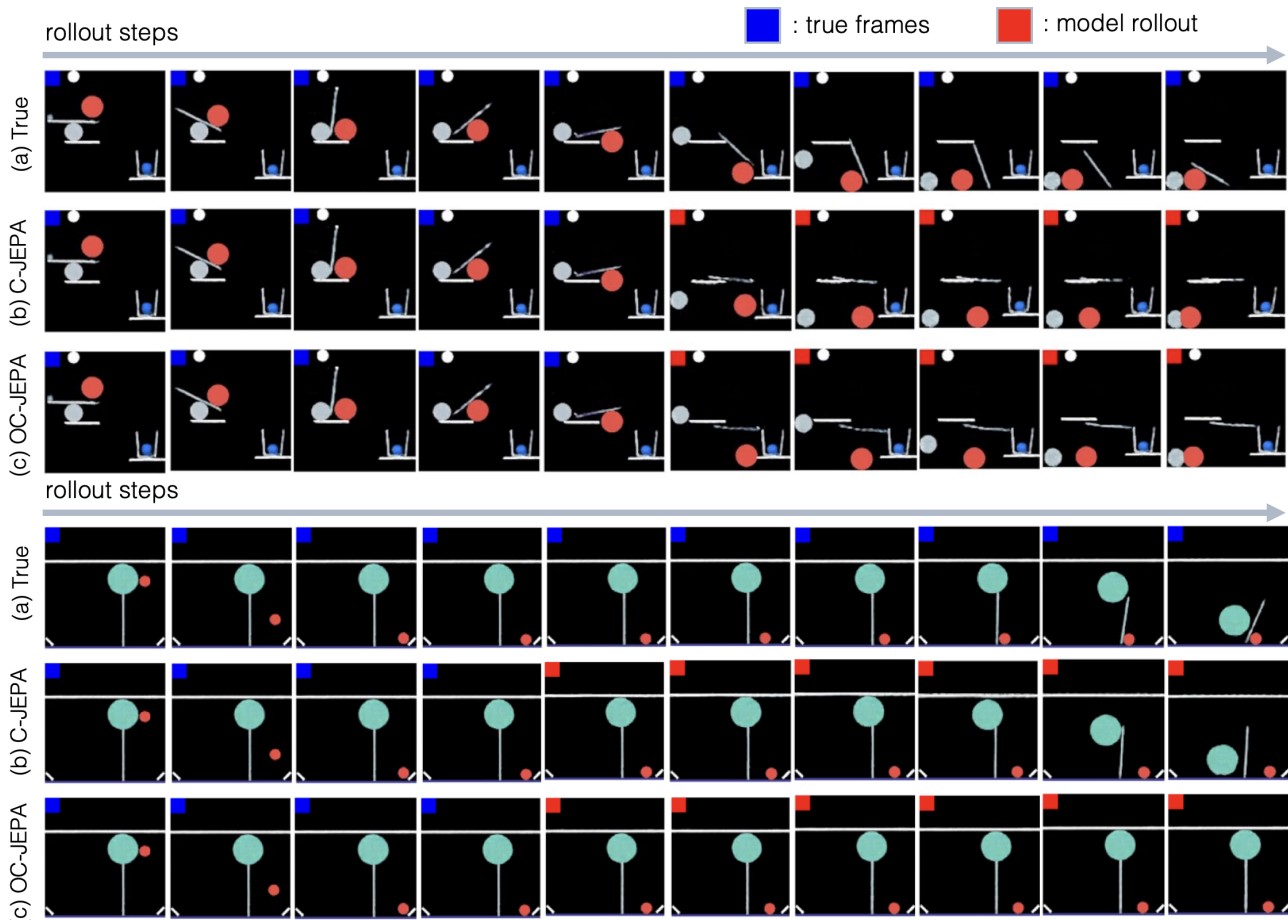

*Figure A3.* Qualitative rollout analysis on PHYRE. We compare imagined future trajectories produced by OC-JEPA and C-JEPA in representative multi-body interaction cases.

- **Token-level masking.** A random subset of individual latent tokens is masked, following standard masked modeling practices. The masking budget is reported as the percentage of masked tokens.

- **Tube-level masking.** Contiguous spatio-temporal tubes of latent tokens are masked, enforcing temporal consistency within masked regions. The masking budget is again reported as the percentage of masked tokens.

For token- and tube-level masking, we match the masking budget to object-level masking by masking an equivalent proportion of latent tokens, and for tube masking we further set the number of masked tubes to correspond to the size of the object masking index set.

**Results.** Tab. A4 compares object-level masking with token- and tube-level masking under matched masking budgets on CLEVRER. Across masking strategies, performance generally improves as the masking budget increases, indicating that masking can act as an effective regularization signal. Rather than viewing object-, token-, and tube-level masking as fundamentally different mechanisms, they can be interpreted as variants that differ primarily in granularity and mask shape. Since masking is applied within a relatively small $T \times N$ latent space in our setting, the performance differences between these strategies are correspondingly limited. However, token- and tube-level masking exhibit higher sensitivity to the masking budget and lead to less consistent improvements. This is because, unlike object-level masking, random masking at the token or tube level can introduce unintended combinations of missing information, such as simultaneously masking multiple objects or all object tokens at a given time step. As a result, the induced prediction task may vary substantially across masking instances, making it harder to consistently enforce interaction-dependent reasoning.

The same trend is reflected in control settings. As shown in Tab. A5 (Push-T), object-level masking maintains robust

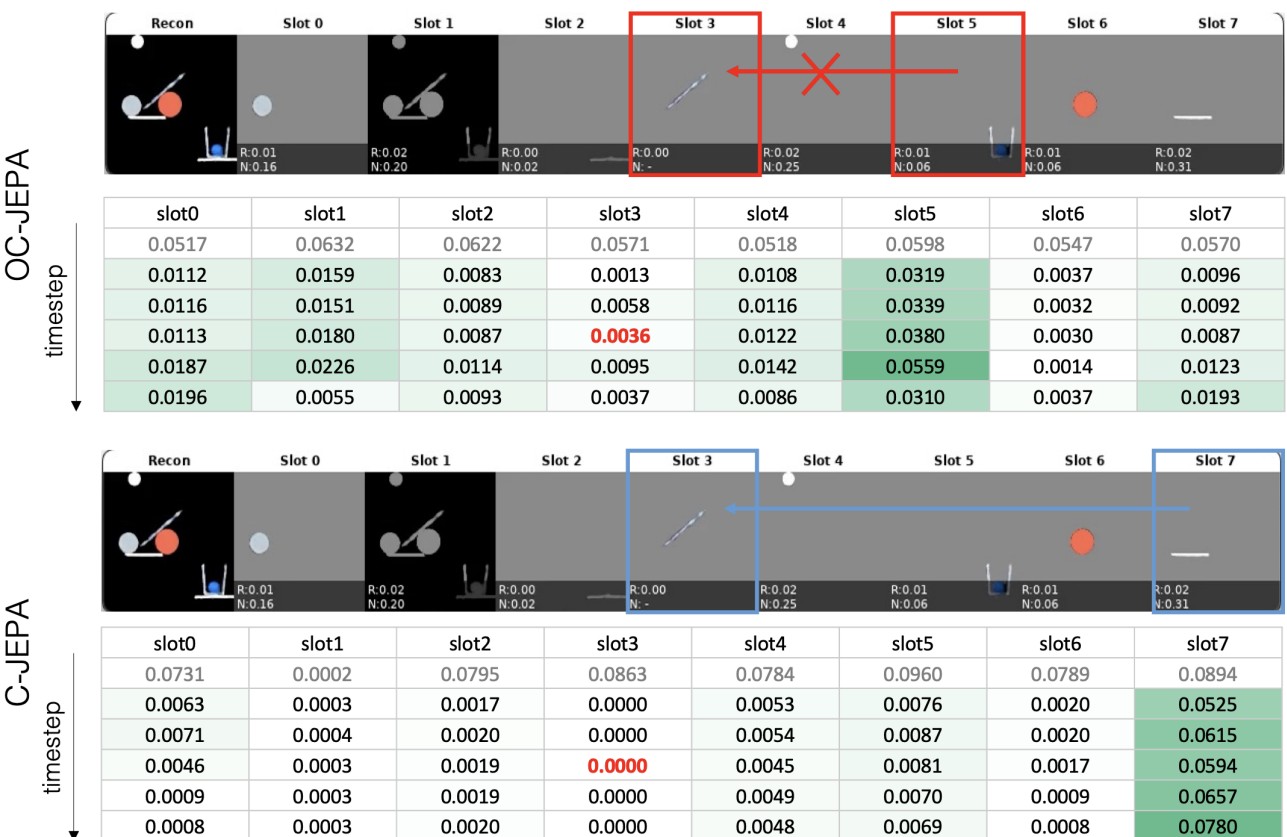

*Figure A4.* Attention-based dependency analysis on PHYRE. We visualize cross-slot attention patterns in the predictor as a qualitative proxy for local temporal interaction structure. Cells marked in red indicate the query token corresponding to the target slot.

planning performance even at higher masking ratios, whereas tube-level masking leads to severe performance degradation under comparable budgets, and token-level masking provides limited or inconsistent benefits. These results indicate that masking entire objects, rather than spatiotemporal tubes or individual latent tokens, is critical for inducing meaningful interaction-aware learning signals without destabilizing prediction or control.

## L. Assumptions

**Assumption 1** (Temporally Directed Predictive Dependencies) *Object-level state transitions are governed by time-directed predictive dependencies. The future state of an object is determined by past object observation and auxiliary variables without requiring instantaneous causal effects within the same time step.*

**Discussion.** This assumption rules out within-timestep causal cycles and provides a well-defined temporal direction for interpreting predictive influence. It is consistent with standard discrete-time modeling, where causal effects are attributed to variables available up to time $t$ and manifest in states at later times.

Importantly, the assumption does not prohibit the predictor from using same-time-step observations during *state completion*. Under object-level masking, the model may condition the reconstruction of a masked token at time $t$ on other objects observed at the same $t$. Such within-step conditioning should be understood as completing missing information under partial observability, rather than positing instantaneous causal generation among objects.

Accordingly, we define *causal influence* operationally through time-lagged predictive dependencies: variables that help predict future states from past observations and auxiliary inputs. This avoids over-interpreting attention weights as instantaneous causal edges while remaining compatible with attention-based predictors that mix information within a window.

*Table A4.* Effect of masking strategies and masking budgets on CLEVRER performance. For token- and tube-level masking, the budget is reported as a percentage of masked tokens. For object-level masking (C-JEPA), the budget is reported as the number of masked objects out of a total of seven.

| Masking Method | $|\mathcal{M}|/7$ | Average per que. (%) | Counterfactual (%) | | Explanatory (%) | | Predictive (%) | | Descriptive (%) |
|---|---|---|---|---|---|---|---|---|---|
| | | | per opt. | per que | per opt. | per que | per opt. | per que | |
| Object | 1/7 | 83.95 | 80.34 | 49.67 | 93.41 | 82.22 | 88.28 | 78.58 | 90.39 |
| | 2/7 | 84.56 | 80.61 | 50.25 | 93.53 | 82.54 | 88.68 | 79.56 | 91.02 |
| | 3/7 | 87.61 | 86.49 | 63.60 | 95.35 | 87.30 | 91.16 | 83.86 | 91.98 |
| | 4/7 | 89.40 | 88.67 | 68.81 | 96.62 | 90.74 | 93.03 | 86.93 | 92.84 |
| Token | 14% | 85.69 | 83.46 | 56.92 | 94.76 | 85.82 | 86.81 | 76.13 | 91.18 |
| | 28% | 87.83 | 87.46 | 65.82 | 95.85 | 88.64 | 89.60 | 80.80 | 91.89 |
| | 42% | 84.43 | 80.69 | 51.08 | 93.77 | 82.87 | 86.46 | 75.54 | 90.90 |
| | 56% | 89.32 | 88.74 | 68.88 | 96.70 | 90.87 | 90.31 | 82.20 | 93.01 |
| Tube | 14% | 86.62 | 84.05 | 57.83 | 94.86 | 86.00 | 88.77 | 79.56 | 92.06 |
| | 28% | 87.87 | 85.42 | 61.71 | 95.67 | 88.43 | 89.18 | 80.01 | 92.74 |
| | 42% | 88.94 | 88.10 | 67.21 | 96.30 | 89.83 | 91.05 | 83.50 | 92.84 |
| | 56% | 89.46 | 89.25 | 69.81 | 97.06 | 91.61 | 90.68 | 82.88 | 92.90 |

*Table A5.* Effect of masking strategies and masking budgets on CLEVRER performance. For token- and tube-level masking, the budget is reported as a percentage of masked tokens. For object-level masking (C-JEPA), the budget is reported as the number of masked objects out of a total of four.

| $|\mathcal{M}|/4$ | Object | Token | Tube |
|---|---|---|---|
| 1/4 (25%) | 88.67% | 84.67% | 55.33% |
| 2/4 (50%) | 82.67% | 84.00% | 5.33% |

**Assumption 2** (Shared Transition Mechanism) *Latent state transitions are governed by a shared mechanism. Although object states vary across time and episodes, the conditional distribution of future states given a finite history remains invariant across trajectories.*

**Discussion.** This is the standard stationarity assumption underlying learned world models: a single predictor is trained to approximate the same conditional dynamics across episodes. It allows pooling experience to learn transferable structure and supports evaluation under rollouts that recombine histories and actions.

The assumption is compatible with diverse trajectories because it constrains only the conditional distribution given history, not the marginal distribution of states. In practice, moderate nonstationarities (e.g., nuisance variations) are typically absorbed by the encoder and model capacity; severe regime shifts would require explicit conditioning on context or environment identifiers, which is outside our scope.

**Assumption 3** (Object-Aligned Latent Representation) *Each slot corresponds to a coherent object-level state variable and these representations provide a sufficient abstraction for reasoning about object dynamics.*

**Discussion.** This assumption is a minimal requirement for object-level interventions to be meaningful: masking a slot should correspond to removing access to an object-level variable rather than an arbitrary mixture of entities. We do not require perfect disentanglement, fixed semantics across scenes, or a one-to-one correspondence with human-defined objects. Instead, we require sufficient *stability* so that slots behave as coherent carriers of object-level information over time, enabling the predictor to treat them as entities for dynamics modeling.

This is consistent with common practice in slot-based perception. At the same time, it clarifies a practical limitation: the achievable performance of object-centric world models is bounded by the fidelity of the underlying encoder, and violations of object alignment can weaken the intended intervention effect of masking.

**Assumption 4** (Finite-History Sufficiency) *A finite history window of length $T_h$ contains sufficient information for predicting*

*future object states under the predictive dependencies of the system.*

**Discussion.** This assumption reflects the operational constraints of learning and planning, where the model conditions on a bounded context window. It is weaker than a first-order Markov assumption because it allows higher-order dynamics that require temporal context (e.g., inferring velocity from multiple frames). In practice, $T_h$ trades off information completeness against computational cost, and the appropriate choice depends on the time scale of interactions and the encoder's ability to summarize motion cues.

When the true system exhibits longer memory than $T_h$, the predictor can still learn an approximation, but accuracy may saturate. Our analysis therefore characterizes the inductive bias induced by masked completion *within* the chosen window, which is the regime relevant to the implemented model.

# M. Theoretical Analysis

## M.1. Proof of Theorem 1

*Proof.* Fix an object index $i$ and a time $t$. Let $Y := z_t^i \in \mathbb{R}^d$ denote the target masked state, and let $X := Z_T^{(-i)}$ denote the observable completion context. Consider any measurable predictor $h$ that maps $X$ to a prediction $\hat{Y} = h(X)$. The (unconditional) MSE risk is

$$\mathcal{R}(h) := \mathbb{E}\big[\|Y - h(X)\|_2^2\big].$$

By the tower property, we can write $\mathcal{R}(h) = \mathbb{E}\big[\mathbb{E}[\|Y - h(X)\|_2^2 \mid X]\big]$. Define $m(X) := \mathbb{E}[Y \mid X]$. Expanding the squared norm and using $\mathbb{E}[Y - m(X) \mid X] = 0$, we obtain the standard conditional risk decomposition:

$$\mathbb{E}\big[\|Y - h(X)\|_2^2 \mid X\big] = \mathbb{E}\big[\|Y - m(X)\|_2^2 \mid X\big] + \|m(X) - h(X)\|_2^2.$$

The first term does not depend on $h$, and the second term is minimized (pointwise in $X$) uniquely by choosing $h(X) = m(X)$ almost surely. Thus the Bayes-optimal predictor for MSE satisfies

$$h^*(X) = \mathbb{E}[Y \mid X] \quad \text{and hence} \quad \hat{z}_t^{i\,*} = \mathbb{E}\Big[z_t^i \mid Z_T^{(-i)}\Big],$$

which gives the first equality in Eq. equation 10.

It remains to show that conditioning on $Z_T^{(-i)}$ is equivalent to conditioning on the influence neighborhood $\mathcal{N}_t(i)$. By Definition 1, $\mathcal{N}_t(i) \subseteq Z_T^{(-i)}$ and

$$p\Big(z_t^i \mid Z_T^{(-i)}\Big) = p\big(z_t^i \mid \mathcal{N}_t(i)\big).$$

Therefore, for any integrable function $\varphi$,

$$\mathbb{E}\Big[\varphi(z_t^i) \mid Z_T^{(-i)}\Big] = \mathbb{E}\big[\varphi(z_t^i) \mid \mathcal{N}_t(i)\big] \quad \text{a.s.}$$

Taking $\varphi$ to be the identity function on $\mathbb{R}^d$ yields

$$\mathbb{E}\Big[z_t^i \mid Z_T^{(-i)}\Big] = \mathbb{E}\big[z_t^i \mid \mathcal{N}_t(i)\big],$$

which gives the second equality in Eq. equation 10.

Finally, let $h$ be any predictor that fails to utilize information in $\mathcal{N}_t(i)$. Then there exists a set of contexts with nonzero probability on which $h(X) \neq h^*(X) = \mathbb{E}[Y \mid X]$, implying $\|m(X) - h(X)\|_2^2 > 0$ on that set. By the decomposition above, $\mathcal{R}(h) > \mathcal{R}(h^*)$, so $h$ cannot attain the minimum achievable expected reconstruction error under masked completion. $\square$

## M.2. Interpretation of Theorem 1.

Theorem 1 characterizes predictive sufficiency under masked observability, rather than recovery of true causal interactions. A predictively sufficient influence neighborhood may align with a structurally meaningful interaction set when the object-centric latents are well aligned with physical entities, the relevant interaction variables are observed within the history window, and spurious correlations are insufficient to predict the masked state. Under latent confounding or partial observability, however, the influence neighborhood may also include variables that are informative without being direct causal parents.

**M.3. Why object-level trajectory masking suppresses shortcuts.**

Complete object-level trajectory masking removes access to the target object's history over the temporal window, while retaining only a minimal identity anchor. This differs from token- or tube-level masking, where nearby observations of the same object may still allow the predictor to recover the missing state through temporal interpolation or self-dynamics. By suppressing these same-object shortcuts, object-level trajectory masking makes contextual information from other objects more important for minimizing the masked prediction loss.

**M.4. Justification of Corollary 1**

We provide a justification for Corollary 1, which states that optimizing the masked prediction objective induces attention patterns aligned with the influence neighborhood.

Consider an attention-based predictor in which each predicted object state is computed by aggregating information from input variables through learned attention weights. Under object-level masking, Theorem 1 implies that variables outside the influence neighborhood $\mathcal{N}_t(i)$ are conditionally uninformative for predicting the masked state $z_t^i$. Consequently, assigning attention to such variables cannot reduce the Bayes risk.

In an expressive attention-based model class, there therefore exist optimal solutions that concentrate attention on variables within $\mathcal{N}_t(i)$ while assigning negligible weight to uninformative variables. Repeated exposure to diverse masking patterns acts as a collection of latent interventions, favoring predictive dependencies that remain stable across interventions. This gives rise to state-dependent and soft relational patterns aligned with the intervention-stable influence neighborhood.

This interpretation is closely related to prior work on invariant causal prediction (Peters et al., 2016) and invariant risk minimization (Arjovsky et al., 2020), which study predictors whose dependencies remain stable across environments or interventions. In C-JEPA, object-level masking plays an analogous role by inducing intervention-stable predictive dependencies in latent space, without explicitly estimating a causal graph.

