# OpenReview forum: "Causal-JEPA: Learning World Models through Object-Level Latent Masking"
_ICML.cc/2026/Conference — ICML 2026 regular_

### Official Review · Reviewer_Zvoi · 2026-03-08

**Soundness:** 3
**Presentation:** 3
**Significance:** 3
**Originality:** 3
**Overall Recommendation:** 5
**Confidence:** 2

**Summary:**

This paper presents Causal-JEPA, an object-centric world model that introduces a causal inductive bias through object-level masking. Built on JEPA, C-JEPA extends masked joint embedding prediction from image patches to object-centric slot representations. The key idea is to mask selected object slots across the history window during training, retaining only a minimal identity anchor, which forces the predictor to infer the masked object's trajectory from interactions with other objects.

**Compliance With Llm Reviewing Policy:**

Affirmed.

**Final Justification:**

The paper provides a theoretically grounded analysis, and the core idea of treating object-level masking as a latent intervention is conceptually clean and well-grounded. The rebuttal has resolved all my concerns.

**Key Questions For Authors:**

- Have the authors evaluated the model under conditions where slot quality is degraded, for example by using a weaker encoder or on videos with heavier occlusion?
- How does C-JEPA perform in more complex settings, such as scenes with dense many-body interactions, significant occlusion, or real-world videos?

**Limitations:**

yes

**Strengths And Weaknesses:**

**Strength**
- The core idea of treating object-level masking as a latent intervention is conceptually clean and well-grounded.
- Achieving comparable planning performance to DINO-WM while using only 1% of the latent token count, and running over 8× faster, is a practically significant result.
- The paper provides a theoretically grounded analysis.

**Weakness**
- Assumption 3 relies on well-formed object-centric representations that slot attention cannot guarantee.
The theoretical analysis assumes each slot corresponds to a coherent, stable object-level variable. However, slot attention is fully unsupervised and prone to slot collapse (multiple slots representing the same object), slot drift (the same object captured by different slots across frames), and background slots mixing with foreground semantics. When this assumption fails, the operation of "masking object i" loses clear semantic meaning, undermining the validity of the theoretical framework.
- Both CLEVRER and Push-T feature simple, sparse interaction structures that are most favorable to C-JEPA. Performance in more challenging settings remains unknown, including scenes with dense many-body interactions, severe occlusions that degrade slot quality, and real-world videos with lighting variation and motion blur.

---

> ### Author Rebuttal · Authors · 2026-03-31
>
> > W1, Q1. Have the authors evaluated the model under conditions where slot quality is degraded?
>
> Thank you for raising this important question. As we noted in the conclusion, performance depends on the quality of the object-centric encoder. Assumption 3 holds only under ideal conditions, and understanding behavior under such sub-optimal conditions is crucial for assessing its robustness. We will add this discussion to the revision.
>
> #### 1. Performance under degraded slot quality
> We have evaluated settings with weaker slot quality, and as expected, performance degrades when the encoder produces less reliable object-centric slots. We view this as a limitation of the current object-centric pipeline rather than a property unique to C-JEPA. When slot representations are less stable or less semantically aligned, object-level masking becomes a noisier supervision signal. In the revision, we will make this dependency more explicit and clarify that C-JEPA operates on top of, rather than replacing, the quality of the slot encoder.
>
> #### 2. Intuition for degradation under weaker slot quality
> We believe there is also a conceptual explanation. Weaker slot quality does not necessarily mean that the representation contains less information overall; more often, the failure mode is that one object is split across multiple slots or that slot identities become temporally unstable. In such cases, a single slot no longer carries a consistent object-level semantic, so masking a slot becomes closer to masking an arbitrary fragment of scene information than to removing one object. While this may still impose some inductive bias, we believe it degrades C-JEPA more strongly because of two factors: (1) the limited number of slots makes it harder to distribute and recover rich relational information once the object-centric decomposition breaks down, and (2) the mask token in C-JEPA is tied to slot identity, so temporal slot instability can inject an incorrect identity signal during prediction.
>
> ---
>
>
> > W2, Q2. How does C-JEPA perform in more complex settings, such as scene with dense many-body interactions?
>
> Thank you for your question. To evaluate C-JEPA in a more complex setting, we conducted experiments on **PHYRE** dataset [1], which requires reasoning about temporally extended physical dependencies such as gravity, momentum transfer, and multi-body collisions, and therefore cannot be solved by simple short-horizon heuristics. We conducted qualitative case studies by
> 1) Rolling out imagined future frames and see how well those rollouts are physically plausible
> 2) Analyzing the attention patterns to understand the learned dependencies.
>
>
> We put visualized samples in this anonymous link: [anonymous.4open.science/r/causal-jepa-phyre-88D3/README.md
> ](https://anonymous.4open.science/r/causal-jepa-phyre-88D3/README.md)
>
> #### 1. Object-level masking improves physical plausibility of rollouts
>
> We observe that OC-JEPA, our baseline without object-level masking, more often produces physically implausible rollouts. In Rollout Case 1, the red ball collides with the green ball in the early frames, so the subsequent fall should be captured; however, the unmasked model fails to track this interaction. In Rollout Case 2, neither model follows the true future exactly, but C-JEPA still produces a rollout that remains physically plausible, while OC-JEPA does not. This suggests that the masking objective remains effective in multi-body settings and helps the model capture more complex physical dependencies.
>
> #### 2. C-JEPA learns more physically meaningful temporal dependencies
>
> Following the perspective of local causal models in SPARTAN [2], we interpret meaningful dependencies as a timestep-specific structure sufficient to explain the next observation, and we use the predictor’s cross-slot attention as a proxy for local temporal interaction structure.
>
> Compared with OC-JEPA, C-JEPA attends to more interpretable and physically relevant slots. For instance, our attention visualization results show that C-JEPA places sharper attention on that relevant slot, whereas OC-JEPA attends more diffusely or to less relevant slots. We do not claim that this constitutes formal causal graph recovery, but we believe it provides useful empirical evidence that the masking objective encourages more physically meaningful, temporally directed dependency modeling in complex environments.
>
> We will include a full description of this experiment, along with additional results and analysis, in the revision. We hope this helps address the concern about performance in more complex settings and provides more direct empirical evidence for our claims about learning temporal dependencies.
>
> ---
>
> #### Reference
>
> [1] PHYRE: A New Benchmark for Physical Reasoning (Bakhtin et al.)
>
> [2] SPARTAN: A Sparse Transformer World Model Attending to What Matters (Lei et al.)

---

> > ### Author Rebuttal · Reviewer_Zvoi · 2026-04-02
> >
> > Thank you for your response. My questions have been resolved.

---

> > > ### Author Response · Authors · 2026-04-05
> > >
> > > Dear Reviewer Zvoi,
> > >
> > > Thank you for taking the time to review our response. We are grateful that your questions have been resolved. We sincerely appreciate your careful reading and helpful feedback, and we will make sure to reflect your comments and suggestions in the revision.

---

### Official Review · Reviewer_ui3V · 2026-03-10

**Soundness:** 2
**Presentation:** 3
**Significance:** 3
**Originality:** 3
**Overall Recommendation:** 4
**Confidence:** 4

**Summary:**

The paper proposes C-JEPA which extends JEPA to incorporate object-centric representations and causal inductive bias. C-JEPA masks objects during training to prevent the model from learning spurious correlations. The paper provides both empirical and theoretical supports for their claims. C-JEPA seems to have the best performance on VQA tasks (CLEVERER), and while it does not beat the baseline on predictive control tasks, it is significantly faster. Theoretical results are provided supporting the idea that masked history prediction forces the model to learn object interactions.

**Compliance With Llm Reviewing Policy:**

Affirmed.

**Final Justification:**

My concerns on writing are addressed. The authors promise to incorporate more nuanced discussion of the results into the paper. I have increased my score from weak reject to weak accept.

**Key Questions For Authors:**

- For Push-T planning, is there other ways to measure success that does not involve ad-hoc thresholding? I'm afraid that the threshold is chosen to favor C-JEPA.
- Why do we not have OCVP-Seq as a another baseline for predictive control on Push-T, given its strong performance on VQA? If not, why?
- How is C-JEPA much better than OCVP-Seq on VQA if the avg accuracy only differs by 0.77%?

**Limitations:**

yes

**Strengths And Weaknesses:**

Strengths:
- The paper is well-written and easy to understand. The authors did a good job at providing intuitions for the proposed metod.
- The method is well-motivated, with both empirical and theoretical support.
-  The result for predictive control does sound impressive; while C-JEPA does not outperform the baseline DINO-WM, it uses a lot less computational resource for inference.

Weaknesses:
- The results for VQA on CLEVERER seems very weak, despite the paper discusses the results as if it is a large gain. The baseline OCVP-Seq gets 83.11% avg accuracy, while C-JEPA gets 83.88% avg accuracy. If the experiments are run with multiple seeds, would the confidence intervals overlap? It doesn't feel right that the paper chooses to ignore to discuss this really close gap of performance. Additionally, in the full results (Table A3), under the descriptive column, C-JEPA is wrongly bold (OCVP-Seq achieves better results).
- With a strong performance of OCVP-Seq on VQA, I wonder if OCVP-Seq can be used for predictive control on Push-T? If not, why? Also for table 3, how is the threshold for success chosen? Is it cherry picked to favor C-JEPA?
- There is a hallucinated reference: the paper cites Bekris et al. 2025 for PushT task. It seems that the paper exists, but the author list is hallucinated -- it should have been Chi et al. 2024. Am I correct?

---

> ### Author Rebuttal · Authors · 2026-03-31
>
> We sincerely thank the reviewer for the feedback and hope these clarifications and planned revisions help address the concerns raised.
>
> > W1, Q3. The results for VQA on CLEVERER seems very weak, despite the paper discusses the results as if it is a large gain. How is C-JEPA much better than OCVP-Seq on VQA if the avg accuracy only differs by 0.77%?
>
> Thank you for this question and we apologize if our wording gave the impression that we were claiming a large numerical leap specifically over OCVP-Seq. That was not our intention. The strong gains emphasized in the paper primarily refer to the substantial improvements over our own baseline, OC-JEPA (Table 1), which more directly isolates the effect of the proposed masking objective.
>
> We also hold OCVP-Seq in high regard as a strong baseline, and our discussion in Section 5.1 already notes its robustness. Our comparison was meant less as a claim of decisive state-of-the-art superiority and more as a conceptual result: Achieving comparable or slightly better performance against a highly optimized model like OCVP-Seq, while operating in a strictly **non-reconstruction setting with a purely decoder-free, latent predictive.** Our results suggest that reconstruction is not necessary for learning robust object-centric temporal dynamics, consistent with [1]. We will revise the text to ensure the tone accurately reflects this nuanced, conceptual comparison.
>
> Our contribution is also better reflected in the counterfactual subset. In Table 2, C-JEPA improves counterfactual accuracy by about 4 absolute percentage points per question over OCVP-Seq, and by about 17 absolute percentage points over the variant without reconstruction loss. We believe this more directly demonstrates the benefit of our masking objective for learning robust temporal dependencies, which is the core contribution of the paper. We will clarify this point in the revision to better reflect the nature of our claims.
>
> ---
>
> > W1. In the full results (Table A3), under the descriptive column, C-JEPA is wrongly bold (OCVP-Seq achieves better results).
>
> We sincerely thank the reviewer for the detailed reading of our paper and for pointing out the typo in Table A3. The bolding for the 'descriptive' column was an error on our part, and we have corrected it to properly highlight in that specific column.
>
> ---
>
> > W2, Q2. I wonder if OCVP-Seq can be used for predictive control on Push-T? If not, why?
>
> Thank you for your question. We would like to clarify the absence of OCVP-Seq in the Push-T experiments. While we highly regard OCVP-Seq as an excellent observation-only video prediction baseline model, performing Model-Predictive Control (MPC) in control tasks like Push-T intrinsically requires the forward model to support action-conditional prediction. Because the standard architecture of OCVP-Seq does not natively support action conditioning, extending it for MPC would require non-trivial, ad-hoc modifications. Therefore, to ensure a rigorous and fair comparison without arbitrarily modifying the authors' original design, we excluded it from the baselines for this specific control task.
>
> ---
>
> > W2. For table 3, how is the threshold for success chosen? Is it cherry picked to favor C-JEPA?
> > Q1. For Push-T planning, is there other ways to measure success that does not involve ad-hoc thresholding? I'm afraid that the threshold is chosen to favor C-JEPA.
>
> In response to your question about the planning task on Push-T, we would like to clarify that we did not introduce any new or task-specific thresholding. Instead, we strictly follow the evaluation protocol of DINO-WM, our primary baseline for the task. In particular, we reuse their exact definition of success, which considers an episode successful when both the translation and rotation errors of the T object fall below fixed thresholds defined in their official implementation (see their codebase: [github.com/gaoyuezhou/dino_wm/env/pusht/pusht_wrapper.py#L66](https://github.com/gaoyuezhou/dino_wm/blob/0a9492fa12044b852ae9e001cc74604b79c8bb0c/env/pusht/pusht_wrapper.py#L66)).
> Therefore, the threshold is not tuned or chosen to favor our method, but directly inherited from prior work, ensuring a fair and consistent comparison.
>
> ---
>
> > W3. There is a hallucinated reference: the paper cites Bekris et al. 2025 for PushT task.
>
> Thank you to the reviewer for identifying this discrepancy. We verified that this error stemmed from the ACM Digital Library metadata. Specifically, the “Export Citation” feature for this article (available at: [dl.acm.org/doi/10.1177/02783649241273668](https://dl.acm.org/doi/10.1177/02783649241273668)) incorrectly attributes the authors and publication year, and we relied on that official metadata when preparing the reference. We will correct the citation in the revision and appreciate the reviewer’s care in catching this issue.
>
> ---
>
> #### Reference
>
> [1] Learning by Reconstruction Produces Uninformative Features For Perception (Balestriero et al.)

---

> > ### Author Rebuttal · Reviewer_ui3V · 2026-03-31
> >
> > My concerns are addressed. Please make sure to revise the paper to incorporate the promised nuanced discussion of the results. I'm raising my score from weak reject to weak accept.

---

> > > ### Author Response · Authors · 2026-04-05
> > >
> > > Dear Reviewer ui3V,
> > >
> > > Thank you very much for your follow-up and for reconsidering your score. We are grateful that our response addressed your concerns. We also sincerely appreciate your note about incorporating the promised nuanced discussion of the results, and we will make sure this is reflected carefully in the revision. Your feedback has been very helpful, and we truly appreciate the time and care you put into reviewing our paper.

---

### Official Review · Reviewer_h577 · 2026-03-11

**Soundness:** 3
**Presentation:** 3
**Significance:** 2
**Originality:** 2
**Overall Recommendation:** 4
**Confidence:** 4

**Summary:**

This paper proposes C-JEPA, an object-centric world model that extends masked joint embedding prediction from image patches to object-centric slot representations. The key design choice is object-level masking during training: selected object slots are suppressed across the history window, forcing the predictor to infer a masked object's trajectory from the evolving states of other objects and auxiliary variables. The authors frame this as a latent intervention with counterfactual-like effects, arguing it induces a causal inductive bias that makes interaction reasoning functionally necessary. A theoretical analysis introduces the notion of influence neighbourhoods and claims to prove that masked history prediction forces the predictor to utilise the minimal sufficient set of contextual variables for recovering a masked object's state. Empirically, the method shows improvements on the CLEVRER counterfactual video QA benchmark and achieves substantial planning efficiency gains on the Push-T manipulation task by operating on a dramatically reduced token space compared to patch-based world models.The paper makes a genuine and useful engineering contribution: object-level masking is a simple, architecturally flexible mechanism that meaningfully improves interaction-aware prediction. However, the theoretical framing requires careful reassessment. In the standard interventionist tradition, an intervention on a variable modifies its structural equation or severs its incoming causal links. Masking an observed slot does neither: it withholds information from the predictor without altering the latent generative process that produces the slot. The influence neighbourhood formalisation, while coherent, characterises predictive sufficiency under observation-level partial information rather than causal structure in the interventional sense. Separating the genuine empirical contribution from the causal framing would make the paper's claims more precise and ultimately more credible.

**Compliance With Llm Reviewing Policy:**

Affirmed.

**Final Justification:**

I have carefully considered the authors’ responses.

They have engaged with all of my questions; however, the issues I raised remain unresolved and, if anything, are further substantiated. While they have outlined potential avenues for improvement, these have not yet been materialized and are therefore difficult to evaluate.

Although this could justify a lower score, I have chosen to maintain my current assessment to provide the authors with a fair opportunity and to acknowledge their openness and sincerity in addressing the feedback.

**Key Questions For Authors:**

My Questions for Authors are:

On the intervention claim: In the paper's causal graph, the latent object state generates the observed slot. Masking the slot withholds it from the predictor but does not modify the latent generative process. Could the authors clarify in what sense this constitutes an intervention on a latent variable, rather than an epistemic operation on an observed downstream quantity? In particular, how does this relate to the standard distinction between hard and soft interventions in structural causal models?

On confounding: If two objects share a latent common cause that drives correlated dynamics, will both appear in each other's influence neighbourhood under C-JEPA's training? If so, what does this imply for the interpretation of influence neighbourhoods as capturing interaction structure rather than shared confounding?

On the theoretical contribution: Could the authors clarify what Theorem 1 establishes beyond the standard result that the conditional expectation is the optimal predictor under mean squared error? Is there a version of the result that provides discriminating insight specific to object-centric representations or to C-JEPA's design?

On masking sensitivity: What determines the optimal masking level, and is there a principled way to select it without access to downstream evaluation? The qualitatively different sensitivity profiles observed across encoders suggest this may be a non-trivial question.

On the counterfactual mechanism: Would a variant that applies simple random slot dropout at the input level, without the JEPA-style latent prediction target, produce similar gains on counterfactual VQA? This would help determine whether the specific training objective structure or simply the exposure to varied relational context is driving the improvements.

**Limitations:**

yes

**Strengths And Weaknesses:**

-----------Strengths-------------

1. The core design idea is simple and reasonable. Object-level masking as a mechanism for encouraging relational reasoning is intuitive and clean. The insight that suppressing an object's observable history forces the predictor to consult peer objects is a meaningful inductive bias, and the paper makes a convincing case that this is qualitatively different from patch-level masked prediction, which optimises for local spatial correlations without requiring object-level interaction reasoning.

2. The OC-JEPA ablation isolates the contribution of the masking objective. The comparison between C-JEPA and its history-unmasked counterpart OC-JEPA uses identical architecture and encoder, differing only in whether history slots are masked. The consistent performance gap across both benchmarks provides evidence that the masking inductive bias is the operative ingredient rather than architectural or representational choices.

3. The efficiency contribution is concrete and practically significant. Operating on object-centric slots rather than image patches reduces the latent token count dramatically. The paper shows this reduction translates directly into substantial planning speedup under model predictive control, which is a genuine practical advance for latent world modelling in control settings.

4. The theoretical framing is honest in important places. Remark 1 explicitly states that the term causal refers to temporally directed predictive dependencies rather than causal identifiability. This is a responsible clarification and the connections drawn to Markov blankets, invariant causal prediction, and invariant risk minimisation are appropriate in spirit. The paper is clearly aware of the distinction between prediction and causation, even if the framing does not always reflect this awareness consistently.

5. The auxiliary variable design is well-motivated. Treating actions and proprioceptive signals as separate entity tokens rather than concatenating them into visual representations preserves the semantic separation between object states and exogenous variables. The ablation in Figure 3 supports this choice empirically, and the reasoning is consistent with principled treatment of observable auxiliary variables in causal representation learning.

-------------Weaknesses--------------
1. Masking an observed slot is not an intervention on a latent variable. This is the paper's most fundamental issue and it is worth engaging with directly and constructively, because resolving it would substantially clarify what the paper actually achieves. In the standard interventionist framework, an intervention on a variable X means either fixing X to a value by severing all incoming edges (hard intervention) or replacing the structural equation governing X with a new mechanism (soft intervention). Both operations act on the variable within the causal graph, modifying its generative process. What C-JEPA does is mask an observed slot, which is a downstream realisation of a latent object state. This replaces the slot value with a mask token in the predictor's input. It does not sever the latent object state from its causal parents. It does not modify the structural equation governing the latent generative process. The underlying data-generating mechanism continues undisturbed. What changes is only the information available to the predictor at training time. This is an epistemic operation on the model's inputs, not an ontological intervention on the causal process. This distinction has a cascading consequence. Because masking does not intervene on the latent cause, the influence neighbourhood that emerges from training reflects predictive sufficiency under observation-level partial information. A latent common cause that simultaneously drives correlated dynamics in two objects will cause each to appear in the other's influence neighbourhood even though neither causally parents the other at the latent level. The influence neighbourhood is therefore a conditional sufficiency concept, not a causal structure concept. The paper would be strengthened considerably by reframing Section 6 around relational inductive bias and predictive sufficiency, which is what the analysis actually proves, rather than causal intervention, which it does not.

2. Theorem 1 is largely a tautology given Definition 1. The theorem proves that the optimal predictor under masked history completion must utilise the influence neighbourhood. But the influence neighbourhood is itself defined as the minimal sufficient subset of contextual variables for recovering the masked object's state. The theorem therefore follows almost directly from the definition of conditional sufficiency and the standard optimality of the conditional expectation under mean squared error. It does not provide discriminating insight specific to C-JEPA, to object-centric representations, or to dynamical systems. Any masked prediction objective over any set of variables satisfies this theorem by construction. The authors might consider whether a more informative theoretical result is available, for instance one that characterises when the learned influence neighbourhoods approximate something structurally meaningful, or one that distinguishes C-JEPA's inductive bias from simpler masking baselines.

3. The masking sensitivity is unexplained and practically important. Performance with one encoder increases monotonically with masking intensity over the tested range, while performance with the other encoder degrades substantially at high masking levels, falling below the unmasked baseline. The paper acknowledges this as an optimal masking regime but provides no principled account of what determines this optimum, no analysis of why the two encoders respond so differently, and no guidance for selecting the masking level without access to downstream task performance. This is a meaningful practical limitation that would benefit from even a qualitative analysis.

4. The empirical evaluation does not fully isolate the claimed mechanism. The paper attributes gains on counterfactual VQA to the interventional structure of object-level masking. However, the more parsimonious explanation is that requiring the predictor to recover object states from peer context encourages learning of relational dynamics that are useful for counterfactual questions, without any specifically interventional mechanism being required. An experiment that compared C-JEPA against a variant using simple input dropout on slots, or random noise injection on slot representations, would help determine whether the specific JEPA-style latent target structure is necessary or whether varied relational training signal is sufficient.

5. Assumption 3 is load-bearing and deserves more scrutiny. The influence neighbourhood analysis requires that each slot corresponds to a coherent object-level state variable. In practice, slot attention can produce unstable bindings across frames, merge multiple objects into a single slot, or split a single object across slots. The paper addresses temporal identity via the identity anchor but does not analyse how slot binding failures propagate through the formalisation or affect the empirical results. A brief empirical analysis of slot stability in the experimental environments, and a discussion of how robustly the method behaves under imperfect segmentation, would substantially strengthen the paper.

---

> ### Author Rebuttal · Authors · 2026-03-31
>
> Thank you very much for your time, insightful observation, and invaluable guidance. We agree that revising Section 6 along the lines you raised will substantially strengthen the paper, and we will do so in the revision.
>
> > W1, Q1. Intervention terminology
>
> We thank the reviewer for highlighting this and acknowledge that our wording was misleading. Your interpretation is correct as we noted in [1]. Although we touched upon this in [1], we recognize that the current wording is imprecise and invites a soft/hard intervention reading, especially given the title *Causal*-JEPA. We will revise the paper to describe masking as controlled information removal inducing a relational inductive bias.
>
> > W1. Influence neighbourhood as predictive sufficiency
>
> The reviewer rightly identifies this connection, as we acknowledged in [2]. We appreciate your comment and we will bring that into the main text explicitly.
>
> ---
>
> > W2, Q3. Better theoretical results
>
> Thank you for this insightful comment. For our current Theorem 1, we will clarify that it is not a structural characterization of true interactions, but a formalization of the basic consequence of object-level masking: under restricted observability, Bayes-optimal prediction depends on a minimal predictively sufficient context.
>
> Following your suggestion, we will expand the discussion in two directions: (1) when a predictively sufficient influence neighborhood may align with a structurally meaningful interaction set, and (2) why complete object-level trajectory masking suppresses same-object shortcut more strongly than other masking strategies. We considered whether these directions could be elevated into new theorems, but we do not believe they would be justified without substantially stronger assumptions and analysis, so we will present these instead as discussion points.
>
> ---
>
> > Q2. Question about confounding effects
>
> Thank you for your question. As we noted in [2], we acknowledge that our framework cannot fully disentangle confounding effects. In the revision, we will make it explicit that influence neighborhoods may reflect both direct interactions and shared confounding.
>
> Admittedly, true interaction structure is generally not identifiable from observations alone in the presence of unobserved confounders. However, we aimed to keep the theoretical framing as close as possible to practical world-modeling settings, even at the cost of providing stronger structural guarantees (L354, right column).
>
> We believe this approach offers a viable alternative to full causal discovery that remains highly effective for reasoning and planning in realistic environments.
>
> ---
>
> > W3, Q4. Masking sensitivity
>
> We thank the reviewer for this constructive feedback. Our intuition is that the safest regime is to mask only one actual foreground object. However, because the number of active foreground objects varies across videos and time while the number of slots is fixed, the same masking ratio can correspond to different semantic corruption levels. We therefore believe the useful regime is determined less by a fixed ratio than by how much effective foreground information is removed while preserving sufficient context, based on dataset statistics. We will discuss more in the appendix.
>
> Also, we hypothesize that the different sensitivity profiles arise from imperfect slot binding and model sensitivity, but we agree that the encoder-specific optimum is not explained by our current analysis. We will mention this as a limitation.
>
> ---
>
> > W4, Q5. Random slot dropout
>
> Thank you for the suggestion. We interpret this question as asking about the masking structure within the same JEPA objective. Accordingly, we compared two masking schemes under the same slot representation and learnable mask token, varying only the masking structure.
>
> - Object-level masking : C-JEPA
> - Token-level masking: Random slot dropout
>
> |  | PushT | PushT | CLEVRER | CLEVRER | CLEVRER | CLEVRER |
> |---|---:|---:|---:|---:|---:|---:|
> | Masking | 25% (1) | 50% (2) | 14% (1) | 28% (2) | 42% (3) | 56% (4) |
> |**object-level** | **88.67** | 82.67 | 83.95 | 84.56 | 87.61 | **89.40** |
> | **token-level** | 84.67 | 84.00 | 85.69 | 87.83 | 84.43 | 89.32 |
>
> On PushT, object-level masking outperforms token-level masking, while on CLEVRER, token-level masking is comparable but less consistent across masking ratios. We believe random slot dropout still provides some relational bias, but object-level masking yields a more stable and meaningful learning signal without destabilizing prediction or control.
>
> ---
>
> #### Reference
>
> [1] Appendix B.1: “object-level masking acts as a **latent intervention on observability**. Rather than intervening on the data-generating process itself, … ”
>
> [2] Appendix B.2: "It may include variables that are correlated through latent confounders, or otherwise informative under partial observability. Influence neighborhoods should therefore be interpreted as **predictively sufficient sets** under masking, ...”

---

> > ### Author Rebuttal · Reviewer_h577 · 2026-04-02
> >
> > I have reviewed the authors’ responses.
> >
> > While they have addressed all of my questions, my concerns remain and, in some cases, are further reinforced rather than resolved.
> >
> > Although this would normally warrant a lower score, I have chosen not to reduce it, in order to give the authors a fair chance and to acknowledge their candid and thoughtful engagement with the feedback.

---

> > > ### Author Response · Authors · 2026-04-05
> > >
> > > Dear Reviewer h577,
> > >
> > > Thank you for taking the time to read our responses closely.
> > >
> > > We understand your concern, and we agree that the current framing was too strong. In particular, we agree that the theoretical framing should be revised so that it more faithfully reflects what is actually established, without leaving room for confusion or overclaiming. Specifically, we will revise the title and main text to remove the intervention terminology and reframe Section 6 accordingly. We will also make explicit in the main text that influence neighborhoods may reflect both direct interactions and shared confounding, and are better understood as a relational inductive bias under restricted observability.
> > >
> > > We also appreciate that your comments helped us separate the paper’s empirical contribution from the stronger causal framing. We will revise the paper accordingly so that its claims are more precise and, we believe, more credible.
> > >
> > > We sincerely appreciate the time and care you put into this review.
> > > Your comments were extremely helpful in clarifying how the paper should be presented, and we will revise it accordingly.

---

### Official Review · Reviewer_2XSa · 2026-03-14

**Soundness:** 3
**Presentation:** 3
**Significance:** 3
**Originality:** 3
**Overall Recommendation:** 4
**Confidence:** 3

**Summary:**

Paper proposes an object-centric JEPA world model that masks selected object slots across the history window while preserving a minimal identity anchor, forcing the predictor to infer an object’s latent state from other objects and auxiliaries rather than relying on self-dynamics or trivial interpolation. The method combines masked history completion with future latent prediction, and the paper argues this induces a causal inductive bias toward interaction reasoning. improves CLEVRER visual reasoning, especially counterfactual questions, and in Push-T planning it achieves near patch-based performance with only 1.02% of the latent tokens and over 8× faster. Also, provides a theoretical argument in terms of influence neighborhoods explaining why masked object prediction should make relational dependencies necessary.

**Compliance With Llm Reviewing Policy:**

Affirmed.

**Key Questions For Authors:**

1. The paper presents an interesting interpretation for causality, but the current evaluation does not compare with ground-truth temporal causal graphs. Say, if such graphs are available, how would you define and evaluate ? Think, what object in the model would be matched to the temporal graph, and what criterion would distinguish recovery of causal structure from learning some dependences only?

**Limitations:**

yes

**Strengths And Weaknesses:**

S:
1. Baseline with 4 different models
2. Same encoder, which is a good call to have a fair comparison 8×faster planning, requiring 673 seconds on average across three seeds to evaluate 50 trajectories, compared to 5,763 seconds for DINO-WM.
3. Largest gains in counterfactual reasoning

W:
1. Causality in the paper title needs a much stronger evidence than what provided. In my very limited understanding, the paper does not show any learned dependencies match the true causal graph of the environment,
2. The patterns recover actual temporal causes
3. The limitations include the statement “while we formally characterize influence neighborhoods, we do not directly validate them on datasets with explicit temporal causal graphs”, so better frame the title as not causal-jepa.

---

> ### Author Rebuttal · Authors · 2026-03-31
>
> > W1, W2, W3. Causality in the paper title needs a much stronger evidence. The paper does not show any learned dependencies.
>
> Thank you for raising this important point. We agree that this issue deserves clearer treatment, and we address it from two angles: first, by clarifying the narrower sense in which we originally used the term causal; and second, by providing new empirical analysis of the dependencies learned by C-JEPA.
>
> #### 1. Clarifying our intended use of “causal”
>
> We would like to clarify the specific context in which we used this terminology. In modern representation learning, including work such as Invariant Causal Prediction [1] and Invariant Risk Minimization [2], "causal" is often used in a broader sense to refer to representations that capture stable, temporally directed predictive dependencies rather than spurious correlations. In this spirit, and as discussed in Remark 1, our objective was designed to encourage robust relational dependencies by blocking identity shortcuts.
>
> #### 2. New empirical analysis
>
> At the same time, we fully agree that the previous version did not provide sufficient empirical support for this framing. To address this more directly, we conducted a new analysis on the PHYRE dataset [3], which requires reasoning about temporally extended physical dependencies such as gravity, momentum transfer, and multi-body collisions. We provide future-frame rollout results and attention-probing visualizations through an anonymous link. Due to the length limit of this response, we kindly refer the reviewer to our answer to Reviewer `Zvoi`, Question 2, for the findings.
>
> ---
>
> > Q1. The current evaluation does not compare with ground-truth temporal causal graphs. If such graphs are available, how would you define and evaluate ? What object in the model would be matched to the temporal graph, and what criterion would distinguish recovery of causal structure from learning some dependences only?
>
>
> Thank you for this thoughtful question. We appreciate the opportunity to clarify this important point.
>
> #### 1. On the absence of ground-truth temporal causal graphs
>
> This is a very fair concern, and in our view it is one of the central challenges in extending causal representation learning and causal discovery to realistic physical environments. In such settings, assuming a single static global interaction graph is often too restrictive. Recent work such as SPARTAN [4] instead emphasizes the importance of learning context-dependent local interaction rather than relying on one fixed global causal graph.
>
> At the same time, we agree with your point that the paper should have provided empirical evidence on such local interaction structure more directly. We hope that the new PHYRE experiment mentioned in our earlier response helps address this concern.
>
> #### 2. What object in the model could be matched to a temporal graph
>
> We agree that defining a graph-like object from the learned model is itself an important question. In our PHYRE analysis, following prior work [4], we use the cross-slot dependency patterns induced by the attention mechanism in the predictor as a proxy for temporal interaction structure. Since ground-truth temporal local causal graphs are not available in PHYRE, we provide qualitative results using this attention-based proxy, following the perspective adopted in [4], and will add examples of these visualizations in the revision.
>
> #### 3. Distinguishing causal recovery from learning dependencies
>
> We also agree that learning predictive dependencies is not the same as recovering the true causal graph. Our current claim is the former, not the latter. More precisely, our goal is to encourage learning of temporally directed predictive dependencies rather than spurious correlations, while acknowledging that the model does not directly identify ground-truth causal structure.
>
> If the question is how one might distinguish true temporal dependencies from merely predictive ones, we believe the most natural evaluation would be in settings with either explicit temporal causal annotations or strong counterfactual supervision. In our current benchmarks, CLEVRER is the closest such setting, since the model is evaluated on counterfactual questions. More broadly, from a world-modeling perspective, the quality of imagined trajectories and downstream planning would be expected to degrade if the learned dependencies were predominantly spurious rather than physically meaningful.
>
> We hope this addresses your question. If you were instead asking about a different notion of graph matching protocol where a true graph is available, we would be happy to clarify further.
>
> ---
>
> #### Reference
>
> [1] Causal inference by using invariant prediction: Identification and confidence intervals (Peters et al.)
>
> [2] Invariant risk minimization (Arjovsky et al.)
>
> [3] PHYRE: A New Benchmark for Physical Reasoning (Bakhtin et al.)
>
> [4] SPARTAN: A Sparse Transformer World Model Attending to What Matters (Lei et al.)

---

> > ### Author Rebuttal · Reviewer_2XSa · 2026-04-02
> >
> > my questions and concerns were answered adequately

---

> > > ### Author Response · Authors · 2026-04-05
> > >
> > > Dear Reviewer 2XSa,
> > >
> > > Thank you very much for your follow-up and for taking the time to review our responses carefully. We are very grateful that you found the concerns to be fully resolved. We sincerely appreciate your thoughtful feedback, and we will make sure the promised clarifications and revisions are fully incorporated into the final version.

---

### Decision · Program_Chairs · 2026-04-30

**Decision:**

Accept (regular)

**Comment:**

This work presents an object-centric world model that extends masked joint embedding prediction from image patches to slot-based object representations. The central design choice is object-level masking during training: specific object slots are suppressed across the temporal context, requiring the model to reconstruct a masked object’s trajectory using the dynamics of other objects and auxiliary signals. This can be interpreted as a latent intervention with counterfactual-like properties thereby introducing a causal inductive bias that makes reasoning about interactions effectively essential.

The paper received 4 reviews with most reviewers praising the writing and the overall idea. There was also a consensus on the soundness of the theoretical framework as well as the efficacy of the experiments. There was a critical question regarding the use of the  term "causal" which I agree with. I think that the authors answer to this is satisfactory although I would still not consider this paper being "causal" in the true sense. Another issue I have with the paper is the experiments which I feel do not do full justice to the framework. Despite this I think this paper is a step in the right direction and thus I recommend weak acceptance.